# BEN-solo factors partition active chromatin to ensure proper gene activation in *Drosophila*

Malin Ueberschär [1], Huazhen Wang[1], Chun Zhang [1,6], Shu Kondo[2], Tsutomu Aoki[3], Paul Schedl[3], Eric C. Lai [4]*, Jiayu Wen[5]* & Qi Dai [1]*

The *Drosophila* genome encodes three BEN-solo proteins including Insensitive (Insv), Elba1 and Elba2 that possess activities in transcriptional repression and chromatin insulation. A fourth protein—Elba3—bridges Elba1 and Elba2 to form an ELBA complex. Here, we report comprehensive investigation of these proteins in *Drosophila* embryos. We assess common and distinct binding sites for Insv and ELBA and their genetic interdependencies. While Elba1 and Elba2 binding generally requires the ELBA complex, Elba3 can associate with chromatin independently of Elba1 and Elba2. We further demonstrate that ELBA collaborates with other insulators to regulate developmental patterning. Finally, we find that adjacent gene pairs separated by an ELBA bound sequence become less differentially expressed in *ELBA* mutants. Transgenic reporters confirm the insulating activity of ELBA- and Insv-bound sites. These findings define ELBA and Insv as general insulator proteins in *Drosophila* and demonstrate the functional importance of insulators to partition transcription units.

[1] Department of Molecular Bioscience, the Wenner-Gren Institute, Stockholm University, Stockholm, Sweden. [2] Laboratory of Invertebrate Genetics, National Institute of Genetics, Mishima, Japan. [3] Department of Molecular Biology, Princeton University, Princeton, NJ, USA. [4] Department of Developmental Biology, Memorial Sloan Kettering Institute, New York, NY, USA. [5] Department of Genome Sciences, The John Curtin School of Medical Research, The Australian National University, Canberra, Australia. [6] Present address: State Key Laboratory of Developmental Biology of Freshwater Fish College of Life Sciences, Hunan Normal University, Changsha, China. *email: laie@mskcc.org; Jiayu.Wen@anu.edu.au; qi.dai@su.se

Proper gene regulation requires coordinated activities of distinct classes of cis- and trans-regulators. Insulators (or boundary elements) are a special type of cis elements that constrain enhancer–promoter interactions[1–5] and set chromatin boundaries[6]. Historically, boundary or enhancer-blocking activities of newly identified insulators were mostly tested on a one-on-one basis in transgenic lines or genetically dissected for individual loci. Recent advances in genomics and chromatin structure capture techniques allowed more systematic identification of insulators and also assigned new properties to them in chromatin architecture organization (reviewed in refs. [7,8]).

The activity of insulator elements depends upon their associated factors. The zinc-finger protein CTCF seems to be the only insulator protein conserved between vertebrates and invertebrates. In addition to its established roles as an insulator in chromatin organization, long-range regulatory element looping and enhancer segregating[7], several of the early studies on mammalian CTCF indicated that it functions in transcriptional repression[9,10]. More than a dozen of proteins have been shown to have insulator function in Drosophila[11]. According to the combinatory co-occupancy patterns of the five insulator proteins CP190, BEAF-32, CTCF, Su(Hw), and Mod(mdg4), Drosophila insulators were divided into two classes[12]. Class I insulators are mainly bound by CP190, BEAF-32, and CTCF in active chromatin regions proximal to promoters, while class II insulators are mostly bound by Su(Hw) located in distal intergenic loci. However, at the functional level, how these factors cooperate remains unclear.

The BEN (BANP, E5R, and NAC1) domain is a recently recognized domain present in a variety of metazoan and viral proteins[13]. Several BEN-containing proteins including mammalian BANP/SMAR1[14,15], NAC1[16,17], BEND3[18], and the C isoform of Drosophila Mod(mdg4)[12,19] have chromatin-associated functions and have been linked to transcriptional silencing. We and others showed that the BEN domain possesses an intrinsic sequence-specific DNA-binding activity. Mammalian RBB, a BEN and BTB domain protein, binds to and directly represses expression of the HDM2 oncogene through interaction with the nucleosome remodeling and deacetylase (NuRD) complex[20]. Drosophila Insv binds to a palindromic motif, TCCAATTGGA and its variants (TCYAATHRGAA), and represses genes in the nervous system[21]. Two other Drosophila BEN proteins, Elba1 and Elba2, along with the adaptor protein Elba3, are assembled in a heterotrimeric complex (ELBA) and associate with the asymmetric site "CCAATAAG" in the Fab-7 insulator[22]. elba1 and elba3 are closely linked in the genome and specifically expressed during the mid-blastula transition, which restricts ELBA activity to this early developmental window. Interestingly, the genes encoding Insv and Elba2 are also arranged next to each other in the genome, even though their gene products show different tissue specificity in later developmental stages.

Most of the BEN-domain proteins contain other characterized motifs. However, Insv, Elba1, Elba2, and several mammalian homologs, such as BEND5 and BEND6, harbor only one BEN domain and lack other known functional domains. Thus, we refer to this sub-class as BEN-solo factors[23,24]. Our previous work demonstrated that Insv and ELBA BEN-solo factors share common properties, e.g., binding to the palindromic sites as homodimers and repressing reporter genes in cultured cells, but also display distinct activities, e.g., Insv being the only one that interacts with Notch signaling and its inability to bind to the asymmetric site[23]. Interestingly, the Fab-7 insulator requires ELBA for its early boundary activity, but also needs Insv in later development[25].

It remains to be determined how the ELBA factors regulate gene expression and embryogenesis. In this study, we have comprehensively characterized the three Drosophila BEN-solo factors and the adapter protein Elba3, by analyzing DNA-binding preferences (symmetric versus asymmetric), chromatin binding inter-dependence (homodimers versus heterotrimeric complex) and mechanisms in gene regulation (repressor versus insulator). Our ChIP-seq (chromatin immunoprecipitation followed by deep sequencing) analyses show that ELBA and Insv bind many common and distinct genomic regions. Unexpectedly, Elba3 associates with chromatin even in the absence of its DNA-binding partners Elba1 and Elba2. Our ChIP-nexus (chromatin immunoprecipitation experiments with nucleotide resolution through exonuclease, unique barcode, and single ligation) assay distinguishes asymmetric heterotrimeric binding pattern of Elba1 and Elba2 from symmetric homodimer pattern of Insv. Although all four factors repress transcription, only the ELBA factors genetically interact with GAF and CP190 and are required for embryonic patterning. Finally, we show that adjacent genes separated by ELBA binding are less differentially expressed in the ELBA mutants. Insv-associated adjacent genes do not show such a global effect, despite individual loci relying on Insv insulation. And ELBA- and Insv-bound elements block enhancer–promoter interaction in transgenic reporters. Collectively, these findings indicate a role of ELBA and Insv as general insulators in partitioning transcription units in Drosophila.

## Results

**The ELBA complex shares many genomic binding sites with Insv.** We previously described genomic binding for Insv whose ChIP-seq peaks cover numerous genomic sites that bear its specific binding motif (CCAATTGG and variants thereof)[24]. A few individual sites in the Fab-7 and the Fab-8 insulators were known genomic locations of the ELBA complex[22,25,26]. We intended to broaden this perspective by generating ChIP-seq data for each of the three ELBA factors from the blastoderm stage of embryos, which covers the peak expression of the ELBA factors[23]. For uniformity of comparing peak-calls, we re-made Insv ChIP-seq data in parallel with the ELBA libraries.

To investigate inter-dependence of factor binding and biological function of the Elba factors, we generated frame-shift mutant alleles for elba1, elba2, and elba3 (Supplementary Fig. 1a). These mutants are viable and fertile, similar to insv mutant. The fluorescent staining confirms that Elba1 and Elba3 are absent from their cognate mutants, while still expressed in the non-cognate mutants. As the Elba2 antibody did not work in immunostaining, we performed RT-qPCR to quantify the mRNA levels of each factor (Supplementary Fig. 1b). The wild-type (wt) mRNAs of each factor are depleted in its cognate mutant, while in the non-cognate mutants the mRNA levels of the ELBA factors are slightly increased. Whether the increase is due to genetic compensation[27] is unclear, and not a focus of this study. We were unable to quantify protein levels of these factors due to technical limitation, so whether an increase in mRNA levels leads to an increase in protein levels remains undetermined. Nevertheless, mutation in any of the ELBA genes does not abolish expression of the other ELBA genes.

We first determined the best suitable normalization control among Input, IgG, and mutant ChIP libraries (Fig. 1a; Supplementary Table 1). The peaks obtained by using the ChIP-seq reads of wild type against those of the cognate mutant showed the highest motif enrichment, compared with the ones against Input or IgG from wt (Supplementary Fig. 1c). Therefore, we used cognate mutant ChIP signal as a negative control for all ChIP-seq peak callings in this study.

We called 3151, 1468, 6525, and 4927 peaks for Elba1, Elba2, Elba3, and Insv, respectively, by using each set of wt ChIP peaks

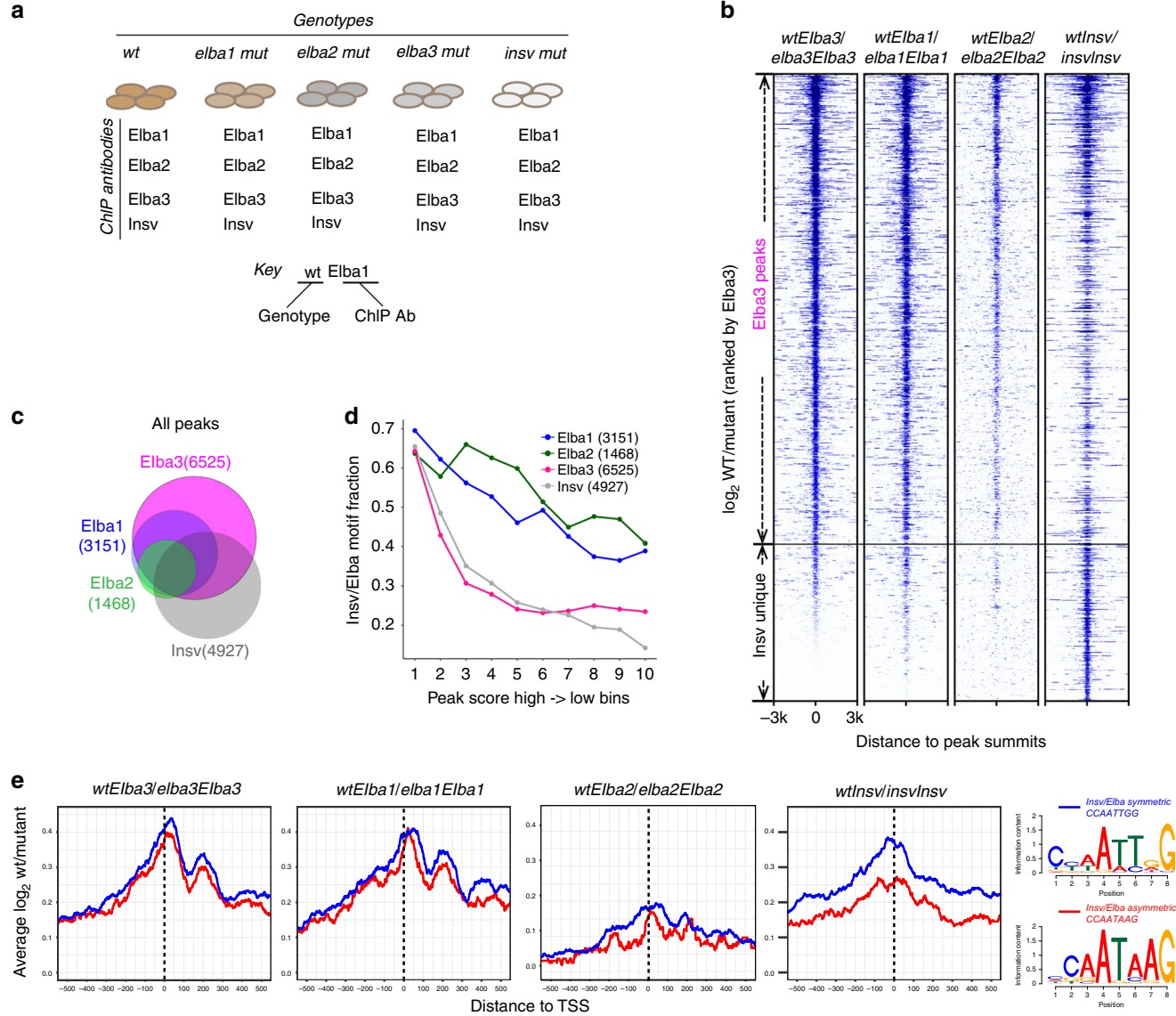

**Fig. 1 Comparison of the ELBA complex with Insv-binding sites and motifs. a** Illustration of the ChIP-seq samples. The naming scheme: the gene names with all the letters in lower case denote genotypes, and the names with the first letter in upper case denote antibodies in ChIP. Two examples are shown. **b** For each factor, the peaks were called by using the ChIP-seq reads of wt against the ChIP reads of its cognate mutant. A heatmap of the ChIP-seq coverage ratio (log2 wt/mutant) centered at peak summits, ranked by the Elba3 signal, showing the peaks of Elba1, Elba2, and Elba3 largely overlap, while Insv has unique peaks. **c** A Venn diagram shows peak overlapping of the four factors. **d** The peak scores and the Elba/Insv motif enrichment are positively correlated. The peak scores were ranked and divided into ten bins (x-axis), and the fraction of motif-containing peaks was calculated for each bin. **e** The coverage ratio (log2 wt/mutant) centered at TSS is shown for the peaks that contain either the symmetric or asymmetric motif. The Elba factors show similar preference at both types of motifs, while Insv signal is higher at the symmetric motif. Logos of the Insv/ELBA motifs from de novo motif discovery are shown on the right.

against each cognate mutant (Fig. 1a–c; Supplementary Table 1). When ranked according to peak scores of the Elba3 ChIP, signals of the three ELBA factors show extensive correlation. Most of the Insv peaks correlate with the ELBA peaks, but a small subset is distinct (Fig. 1b). The overlapping analysis in the Venn diagram demonstrates that Elba1 and Elba3 peaks cover nearly all of the Elba2 peaks, while about half of the Insv peaks are unique (Fig. 1c). To confirm the signal specificity, we performed ChIP-qPCR with another set of ELBA and Insv antibodies than the ones used in the ChIP-seq experiments in three exemplary loci (Supplementary Fig. 2a–c). *CG12811* represents four-factor binding, *mRpS24* is bound by the ELBA complex, and *kirre/Notch* is uniquely bound by Insv. In all of the cases, the enrichment of ELBA and Insv is consistent with the ChIP-seq

analyses. Importantly, the control pre-immune serum pulled down little DNA, the four specific antibodies did not precipitate a negative region in the gene *CG34245*, and the enrichment is gone in the corresponding cognate mutants.

Our previous study showed that all three BEN-solo factors can bind the symmetric site CCAATTGG, while only the ELBA complex binds the asymmetric CCAATAAG in cell culture[23]. To examine site preference of these factors in the embryo, we performed de novo motif discovery from the ChIP-seq peaks. The known symmetric and asymmetric sites were enriched for all four factors (Fig. 1d, e). Motif occurrence frequency positively correlates with peak scores, with 60–70% motifs in the top 10% peaks for each factor (Fig. 1d). The coverage ratios (Log2 wt/mutant) of the peaks containing the two types of motifs do not

differ between the motif types for the ELBA factors, whereas for Insv the value of the symmetric motif is higher (Fig. 1e). These analyses confirm that the ELBA factors target both types of sites, while Insv favors the symmetric ones.

**Elba3 targets genomic locations beyond the ELBA sites.** Elba3 is the adaptor that bridges the DNA-binding partners Elba1 and Elba2 in ELBA. Surprisingly, many more ChIP-seq peaks are called for Elba3 than for Elba1 and Elba2 (6525 versus 3151 and 1468) (Fig. 1c). This raises the question whether such sites are bound by Elba3 but not by the ELBA complex, or pulled down by the Elba3 antibody but not by the Elba1 or Elba2 antibodies due to technical reasons, such as antibody affinity or epitope configuration. The efficiency of the Elba2 antibody is poor, which may partly account for its fewer ChIP-seq peaks. The Elba1 and the Elba3 antibodies are generally comparable (Supplementary Fig. 1a).

We decided to use two strategies to assess whether there are sites bound only by Elba3, but not by Elba1 and Elba2. First, we examined the properties of the Elba3 sites. We compared five subsets of regions identified from the ChIP-seq analyses on their ChIP-seq signal (wt/mutant ratio) (Supplementary Fig. 1d) and motif enrichment (Supplementary Fig. 1e). These subsets include: (i) the Elba3-unique sites, (ii) the sites bound by all four factors, (iii) the sites with the Elba factors without Insv co-binding, (iv) the sites with Elba3 and Insv without Elba1 and Elba2 binding, and (v) the Insv-unique sites. The four-factor overlapping sites show the highest average coverage and motif enrichment, followed by those co-bound by the Elba factors and those by Elba3 and Insv. The Insv-unique and the Elba3-unique sites have the lowest signal. To validate binding specificity of Elba3 on these low-affinity sites, we tested an exemplary locus (*Mesh1*) by ChIP-qPCR using a different Elba3 antibody than the one used in ChIP-seq (Supplementary Fig. 2d). Consistent with the ChIP-seq result, Elba3 is enriched in *Mesh1* in wt, *elba1*, *elba2*, and *insv* mutants, confirming that Elba3 binding to some of its target sites does not require Elba1 and Elba2.

Second, we tested whether the Elba3 protein exists outside ELBA, using a sequential depletion co-immunoprecipitation experiment. We applied a control IgG, an Elba1, and an Elba3 antibody in the first round of immunoprecipitation to deplete all the Elba1- or the Elba3-containing complexes, and then ran a second round of immunoprecipitation from the supernatant material using the Elba1 or the Elba3 antibodies again (Supplementary Fig. 2e). If all of the Elba3 molecules associate with all of the Elba1 proteins, the Elba3 antibodies should pull down all of the Elba1 proteins and vice versa. As expected, these antibodies depleted nearly all of the corresponding proteins in the first round, as little remained in the second round (Supplementary Fig. 2e, lanes 2 and 8). The Elba3 antibody depleted Elba1 as much as the Elba1 antibody did (compare lanes 2 and 4). However, there is more Elba3 protein left in the second round after the first round of depletion with the Elba1 antibody, compared with that with the Elba3 antibody (lanes 6 and 8). This result demonstrates that there are indeed extra Elba3 protein molecules in addition to the ones associated with ELBA.

**Elba3 is essential for ELBA to target chromatin.** The three ELBA subunits rely on one another to bind the ELBA site in the *Fab-7* insulator in vitro[22]. To assess genetic interdependencies of binding among Insv and the ELBA factors, we generated ChIP-seq data for each factor from the non-cognate mutant background (Figs. 1a, 2a; Supplementary Table 1). Insv binding was not affected by any of the *ELBA* mutations, or vice versa (Fig. 2a; Supplementary Fig. 3a, b). Binding of Elba1 or Elba2 was nearly eliminated in the *elba3* mutant (with only 68 Elba2 peaks

remaining), indicating that Elba3 is an essential component for the endogenous ELBA complex to bind the genome. This was unexpected, given that Elba1 and Elba2 under ectopic expression can form homodimers in cultured cells[23]. Most of the Elba1 sites are lost in the *elba2* mutant with 712 (~20%) peaks left. Nearly all of the Elba2 sites are lost in the *elba1* mutant, with 48 (~3%) remaining. As the expression of the ELBA factors is not reduced in the non-cognate mutant condition (Supplementary Fig. 1a, b), this result suggests that Elba1 and Elba2 targeting to the genome relies on the formation of the ELBA complex.

Notably, Elba3 maintains about half of its peaks in the *elba1* or the *elba2* mutant (assigned as Elba1/2-independent sites) (Fig. 2a, b). We confirmed this using ChIP-qPCR assays with a different set of antibodies in two loci, *Sppl* and *r*, that exemplify the Elba1/2-dependent and independent Elba3 sites, respectively (Fig. 2c, d). The enrichment of the ELBA factors in the *Sppl* locus is gone in all of the *ELBA* mutants, while Elba3 retains its binding to the *r* locus in *elba1*, *elba2*, and *insv* mutants. Notably, the enrichment of Elba3 appears lower in the *elba1* and the *elba2* mutants, suggesting that Elba1 and Elba2 are dispensable for but could stabilize Elba3 binding.

Since Elba3 has no DNA-binding domain, we asked whether Insv mediates Elba3 to the genome in the absence of Elba1 and Elba2. The overlapping fraction of Elba3 and Insv peaks is twofold higher in the Elba1/2-independent sites (50%) than that of the dependent ones (25%). However, half of the Elba1/Elba2-independent peaks do not overlap with Insv peaks (Supplementary Fig. 3d), suggesting that Insv may contribute to or enhance Elba3 binding to some, but not all Elba1/2-independent Elba3 loci.

We assessed other aspects of properties of these two groups of sites. The Elba1/2-dependent sites are more enriched in introns, exons, and distal regions and have a higher frequency of motif occurrence, whereas the Elba1/2-independent sites are mostly at promoter-TSS proximal regions and less enriched for the motifs (Supplementary Fig. 3c, d). The Elba1/2-independent sites have a higher Elba3 ChIP signal coverage (Supplementary Fig. 3e), indicating stronger binding of Elba3 to these sites. Thus, there are intrinsic differences between these two groups of Elba3 sites.

We conclude that Elba3 can bind the genome in three ways, through the ELBA complex, through protein–protein interaction with another DNA-binding factor but still within the ELBA complex, and through protein–protein interaction with another factor without the presence of Elba1 and Elba2 (Fig. 2e).

There are 712 Elba1 peaks remaining in the *elba2* mutant (Fig. 2a; Supplementary Fig. 3f). Among these peaks, 496 of 712 (70%) do not overlap with the Elba1 peaks in wt but with the Elba3 peaks in the *elba2* mutant, raising a possibility that Elba1 has shifted or enhanced its binding to the new loci with Elba3. Compared with the Elba1 peaks in wt, the Elba1 peaks in the *elba2* mutant are less enriched for the Insv/Elba motifs, locate more often in promoter-proximal regions (Supplementary Fig. 3g) and have comparable ChIP-seq coverage (Supplementary Fig. 3i). We compared Insv and Elba1 overlapping fractions in wt with those in the *elba2* mutant, and found no significant difference between these two conditions. This result argues that Insv does not contribute to Elba1 binding to these new genomic loci (Supplementary Fig. 3h).

Thus, the ChIP-seq analyses revealed unexpected in vivo binding properties of the three ELBA factors, in which Elba3 is the essential component and has the ability of targeting genomic sites independent of Elba1 and Elba2.

**ChIP-Nexus differentiates DNA-binding symmetry.** We reported that all the three BEN-solo proteins could bind to the

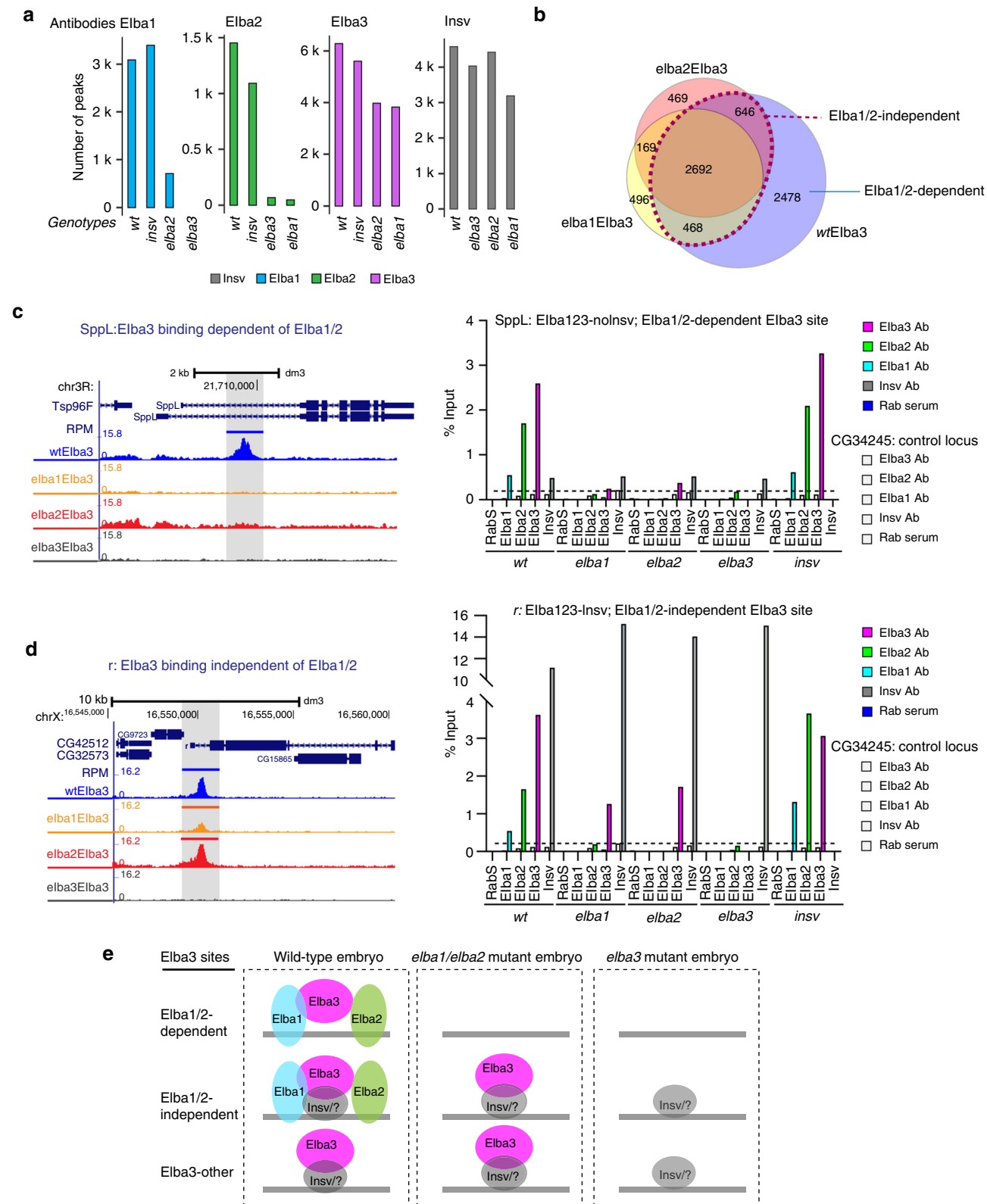

symmetric site as homodimers when overexpressed in cultured cells, while the ELBA complex has higher affinity to the asymmetric site[22,23,25]. Our ChIP-seq analyses suggest that ELBA and Insv associate with both types of sites in the genome (Fig. 1e). Given that ELBA and Insv overlap extensively (Fig. 1) and relatively broad ChIP-seq peaks may limit the resolution of closely spaced factors, we performed ChIP-nexus[28] to better discriminate

binding preference of ELBA and Insv using the same sets of antibodies and the same stage of wt embryos.

One challenge in ChIP-nexus data analyses is the lack of a negative control, because ChIP-nexus from mutant or IgG immunoprecipitations could not produce sufficient material for deep sequencing. This is perhaps due to its inherently low background, as no negative control in ChIP-nexus analysis has

**Fig. 2 Elba3-binding sites are partially independent of the ELBA trimeric complex. a** The ChIP-seq peaks of wt and the non-cognate mutants for each factor were called by using the corresponding ChIP-seq reads against the reads in its cognate mutant. The number of peaks is shown for each factor in each genotype. **b** Overlaps of Elba3 binding in the three genotypes, wt, elba1, and elba2, showing that ~50% wtElba3 sites remain in elba1 and elba2 and are denoted as Elba1/2-independent sites. The wtElba3 sites lost in elba1/2 are Elba1/2-dependent. **c** ChIP-seq tracks of an exemplary locus SppL, of Elba1/2-dependent sites, are shown on the left. On the right, ChIP-qPCR with a second set of ELBA and Insv antibodies confirms specific enrichment of ELBA in wt and insv mutant embryo, but not in any of the elba mutants. Average value from three technical replicates is plotted. A negative control locus CG34245 was examined in parallel. **d** An exemplary locus, r, of Elba1/2-independent Elba3 sites. ChIP-qPCR validation using a second set of antibodies against these four factors confirms the ChIP-seq result. **e** Illustration of three contexts where Elba3 locates to the genome: Elba1/2-dependent, Elba1/2-independent, and other Elba3-binding sites. Source data of the raw qPCR value is available in a Source Data file.

been reported[28,29]. To overcome this problem, we manually spotted coverage intensity and set a stringent cutoff (FDR < 1E-10 for Elba1, Elba3, and Insv, and FDR < 1E-5 for Elba2) according to the signal to background ratio. After applying this cutoff, we defined ChIP-nexus peaks and compared motif occurrence frequencies between the ChIP-seq and ChIP-nexus data sets. ChIP-nexus increased the motif enrichment frequency for Elba3, but not for the other three factors (Supplementary Fig. 4a), while the overlapping fractions of the ChIP-seq and ChIP-nexus peaks show a slightly higher motif enrichment for all the factors (Supplementary Fig. 4b, c). We plotted motif coverage centered at the peak summits. Compared with the ChIP-seq data, the ChIP-nexus peaks have a more centered distribution for both types of consensus sequences (Fig. 3a). To assess whether ChIP-nexus can provide a higher resolution between closely spaced factors, we performed overlapping analyses by allowing maximum peak summit distance to be 10, 25, and 50 nucleotides (nts). When the distance is within 10 or 25 nts, the ChIP-nexus peaks already show more overlapping between the ELBA factors (Supplementary Fig. 4d), whereas the ChIP-seq peaks do not have such a trend. When the distance is 50 nts, the two methods start to show a similar pattern where the ELBA factors overlap extensively while a subset of Insv peaks are separated. Together, the result confirms that ChIP-nexus can achieve high resolution when high stringent peak calling cutoff is applied.

The Fab-7 insulator is one of the top-bound regions by Insv[23] and was the first locus where ELBA was detected[22]. ELBA and Insv both contribute to the function of Fab-7[22,25]. This 1.2 kilobase fragment contains one Insv/ELBA asymmetric and two symmetric sites. While ELBA binds to all three sites in vitro[22,25], Insv has much higher affinity to the symmetric sites[23,25]. The ChIP-seq peaks generally confirm the specificity of ELBA and Insv binding to their expected sites, despite broad peaks overlapping in the entire region (Fig. 3b). The ChIP-nexus peaks are sharp, with a high peak of Elba1 and Elba2 at the ELBA site. Notably, the Elba1 and Elba2 peaks display strand asymmetry: the Elba1 reads primarily cover the "+" strand while the Elba2 reads cover the "−" strand of the CCAATAAG sequence. In contrast, the Elba3 peaks are symmetric. Many other individual loci bound by Elba1 and Elba2 display a similar pattern, as exemplified by the Parp1 locus: Elba1 and Elba2 show strand preference in DNA binding (Fig. 3c).

These observations prompted us to examine binding symmetry at a global level. We calculated the orientation index (OI) for the top 500 ChIP-nexus peaks that contain the Insv/ELBA motifs. The OI value is determined as the ratio of the number of reads from the dominant strand to the total number of reads from both strands. Elba3 and Insv peaks are symmetric at a global level as their OIs are mostly close to 0.5 (Fig. 3d). In contrast, the OIs of Elba1 and Elba2 peaks have a trend toward 1.0, indicating an asymmetric binding pattern. Thus, the ChIP-nexus assay helped to distinguish heterotrimeric binding from homodimer binding (illustrated in Fig. 3e). Despite having similar DNA-binding

domains and expression patterns in the early embryo, Insv displays distinct binding preferences compared to ELBA.

**The ELBA factors repress target gene expression in the embryo.** We previously reported that all three BEN-solo factors possess repressive function: Insv represses neural genes in the embryo, and Insv, Elba1, and Elba2 can all repress reporter expression in cultured cells[21,23]. As Elba3 can target chromatin independent of Elba1 and Elba2, we asked whether Elba3 can repress transcription independent of Elba1 and Elba2. To address this, we tethered Elba3 with the Tet repressor DNA-binding domain (TetR-Elba3) and examined its activity in Drosophila S2 cells on a luciferase reporter driven by an actin enhancer and the tet operator sites. Elba3 represses reporter expression with similar efficiency as the other three proteins (Fig. 4a). Since S2 cells lack Elba1, Elba2, and Insv, this result suggests that the repression activity of Elba3 does not rely on any of the BEN-solo factors.

To investigate how ELBA regulates gene expression in the embryo, we determined gene expression changes between 2–4 h wt and mutant embryos using RNA-seq analysis. We examined fold change for the target genes identified by ChIP-seq, using a gene set enrichment testing (see the Methods section) for the targets as a set (FDR < 1E-5 for Elba1/2/3 and FDR < 0.01 for Insv). While the target genes all display a trend of de-repression, the top-bound genes with the Insv/ELBA motifs show the highest fold change, and the genes with the motifs have higher fold change than those without the motifs (Fig. 4b). This result demonstrates that Insv and ELBA repress their direct targets in the embryo.

We further performed overlapping analysis on the genes that changed expression in the mutants (FDR < 0.2 and FC > 1.5-fold) (Supplementary Fig. 5a, b). There are few downregulated genes shared by the three ELBA mutants. However, the overlapping pattern of the upregulated genes resembles their ChIP-seq peak overlapping pattern (Fig. 1b): the upregulated targets in the elba2 mutant are shared in the elba1 mutant, which are further shared in the elba3 mutant. This suggests that Elba3 binds to and regulates more genes than Elba1 and Elba2. The upregulated genes in the insv mutant partially overlap with those in the elba mutants, suggesting that Insv and ELBA regulate a subset of common targets.

We next asked how each category of target genes, including those bound by the four factors, the ELBA complex, Elba3, and Insv, Elba3 alone, or Insv alone (Supplementary Fig. 1d, e), respond to the four mutant conditions. The genes co-bound by the four factors became significantly upregulated in all the mutants (Supplementary Fig. 5c), with the highest de-repression in the elba3 mutant. The genes bound by the ELBA complex but not Insv displayed mild upregulation in the elba2 mutant, while the genes in the other categories do not change significantly. As the ChIP-seq peaks bound by all four factors have the highest ChIP-seq coverage, expression of the genes associated with these peaks may be most sensitive to loss of ELBA or Insv.

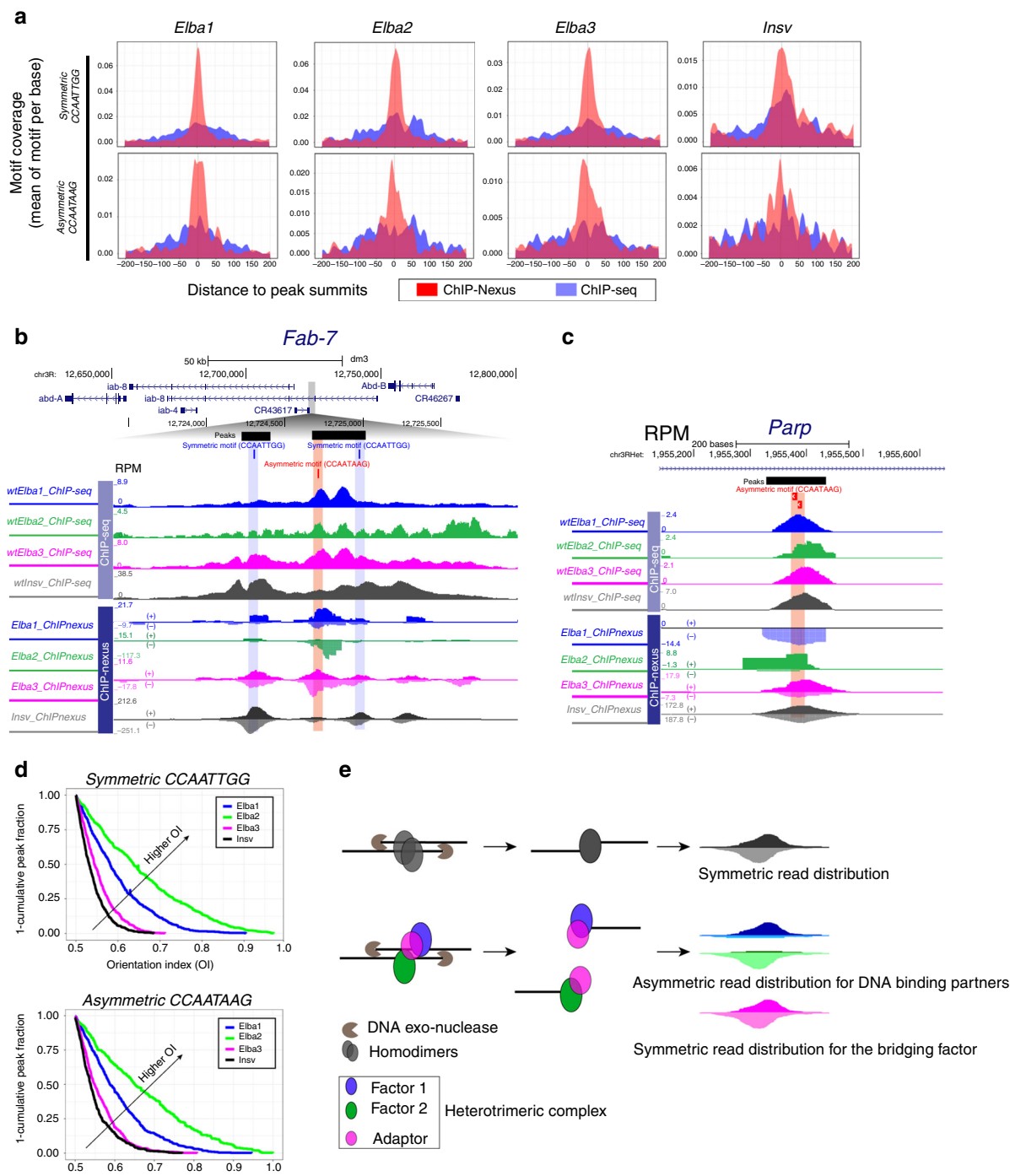

**Fig. 3 ChIP-nexus distinguishes homodimer from heterotrimeric binding. a** Comparison of ChIP-seq and ChIP-nexus, showing centered motif distribution around the peak summits. The motif occurrence centered at the peak summits (x-axis) is plotted with mean motif coverage in "Motif Per Base" on the y-axis. **b** ChIP-seq and ChIP-nexus tracks of the *Fab-7* region show broader ChIP-seq peaks while sharp ChIP-nexus peaks. Note: ChIP-nexus also shows asymmetric read distribution of Elba1 and Elba2 and symmetric distribution of Elba3 and Insv. Elba1 and Elba2 respectively prefer the "+" and the "−" strand of the CCAATAAG motif. **c** Another exemplary locus, *Parp*, exhibits a similar bias of strand asymmetry. **d** The orientation indexes (OIs), ranging from 0.5–1, were calculated for the top 500 ChIP-nexus peaks containing the symmetric or asymmetric motifs (see the Methods section). Elba1 and Elba2 display a higher OI tendency than Elba3 and Insv, suggesting asymmetric binding. **e** Illustration of how ChIP-nexus can capture symmetric versus asymmetric binding patterns by homodimers versus a heterotrimeric complex.

**ELBA is required for *Drosophila* embryonic patterning.** Next, we sought to identify co-factors that work with ELBA and Insv. The highly enriched motifs in the ELBA or Insv ChIP peaks also include the binding sites for three known insulator proteins, CP190, BEAF-32, and GAF (Fig. 5a). These motifs are found in all five categories of binding regions (Supplementary Fig. 1d, Supplementary Fig. 6a). We performed pairwise comparison for the Insv and the ELBA ChIP-seq peaks with the ChIP–Chip peaks of CP190, BEAF-32, CTCF, GAF, Mod(Mdg4), and Su(Hw) (modEncode data sets). Consistent with our previous analysis[23],

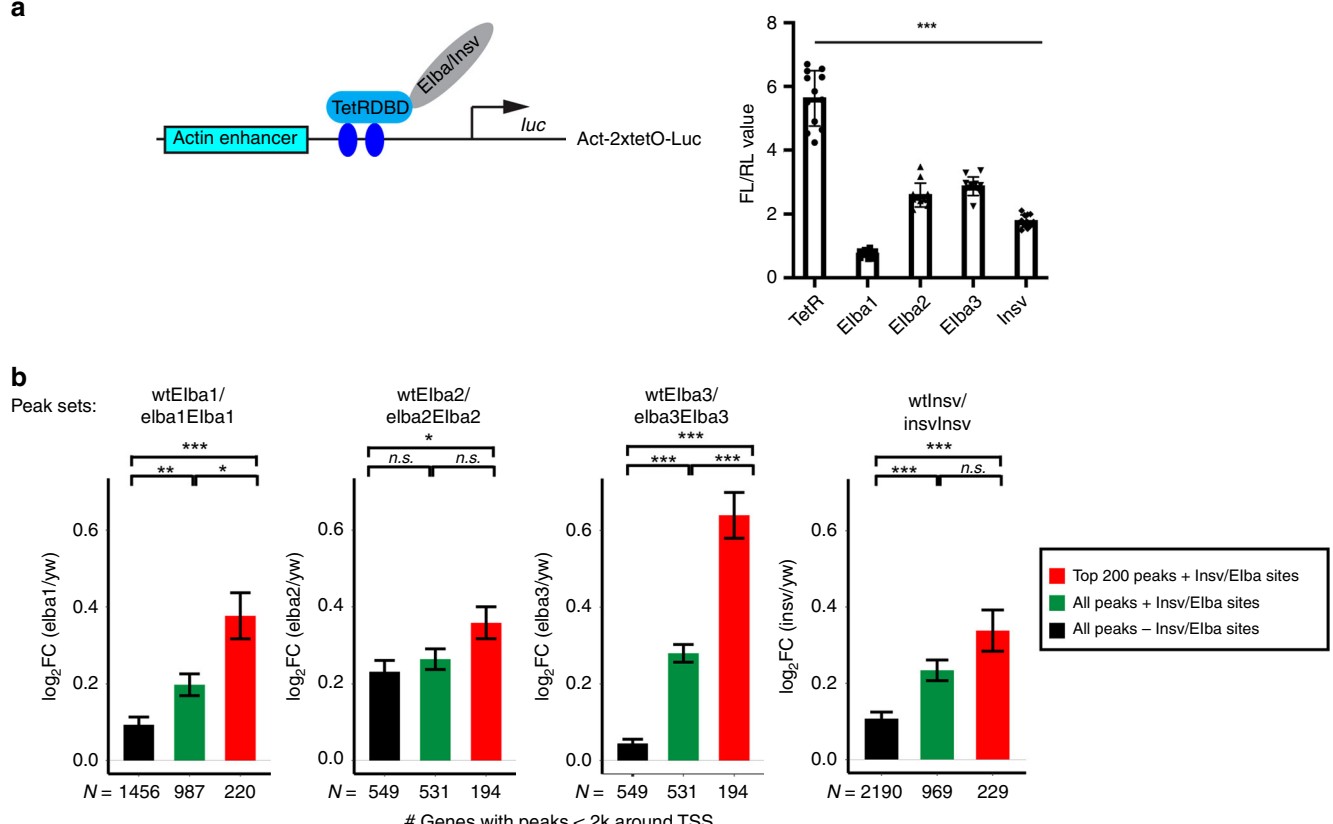

**Fig. 4 Elba and Insv repress target gene expression. a** Luciferase reporter assays with the TetR-DNA-binding domain (DBD) fusion proteins. The Elba and Insv proteins were fused with the TetR-DBD and thus brought to the 2 × *Tet* operator sites in the luciferase reporter. The values of firefly/renilla luciferase ratio are plotted and represent the average from three biological replicates with each biological replicate having four technical replicates. So 12 values are included for each condition ($n = 12$), and error bars represent S.D. *p*-value is calculated using a Student's *t* test, two-tailed, type 1. ***$p < 0.001$. Source data are available in a Source Data file. **b** The four bar plots show log$_2$ fold change of RNA level of target genes in the mutant versus wt based on RNA-seq data. For each factor, the top 200 bound genes that contain the Elba/Insv motifs (red) were more upregulated than all the target genes with the Insv/Elba motifs (green) and the target genes without the Insv/Elba motifs (black). Pairwise statistical significance among the three sets were calculated by two-tailed *t* tests, and *p*-values were adjusted by the Bonferroni multiple testing correction method (*$p < 0.05$, **$p < 0.01$, ***$p < 0.001$, n.s. not significant).

Insv shares many binding regions with CP190, BEAF-32, CTCF, and Mod(Mdg4), and fewer with GAF and Su(Hw). The ELBA factors display similar co-occupancy patterns (Fig. 5b), suggesting that they mainly associate with class I insulators.

Knockdown of ELBA in the early embryo was shown to influence boundary activity of the HS1 element in the *Fab-7* insulator[22]. Loss of ELBA or Insv also influences gene expression (Fig. 4). However, the *ELBA* and *insv* mutants are viable and do not display obvious morphologic defects. We reasoned that this could be due to redundancy with other insulator factors as they co-occupy similar genomic locations. To test this hypothesis, we examined genetic interactions between *ELBA* or *insv* and *GAF* or *CP190*. We crossed a null allele of *GAF*, *Trl^{R85}* [30], a hypomorphic allele of *GAF*, *Trl^{13C}* [31], and a null allele of *CP190*, *CP190^{P11}* [32], into the background of *elba1*, *elba2*, *elba3*, or *insv* homozygous mutant backgrounds, and scored for synthetic adult lethality (Supplementary Table 2) and defects in embryonic patterning (Fig. 5c, Table 1). It was shown that *insv* genetically interacts with *GAF* in the function of *Fab-7*[25], and that the Insv protein physically interacts with CP190[23,33]. However, we did not observe interactions between *insv* and *GAF* or *CP190* with respect to viability or embryonic patterning. It is conceivable that Insv and these factors work together in other developmental contexts. In contrast, animals homozygous for *elba3* or *elba2* in combination with heterozygous *Trl^{R85}* do not survive to adulthood. Importantly, the lethality of *elba2* and *Trl^{R85}*

double-mutant is fully rescued by a pBAC transgene expressing the endogenous level of *elba2* (Supplementary Table 2). In the combinations with heterozygous *CP190^{P11}*, homozygous *elba1* and *elba3* mutants are lethal, suggesting distinct involvement of the three ELBA subunits with other insulator proteins in developmental processes.

A fraction of embryos mutant for *ELBA* and *Trl^{R85}* or *Trl^{13C}* also displayed severe embryonic patterning defects such as disrupted denticles and head involution, with the *elba3* and *Trl^{R85}* combination having the strongest effect (only 4% normal looking embryos) (Fig. 5c, Table 1). Embryos of the *elba2* and *Trl^{R85}* combination did not show patterning defect, presumably due to maternal contribution from *elba2* heterozygous mothers. Indeed, when embryos were produced from homozygous *elba2* and heterozygous *Trl^{13C}* females, a fraction of them displayed defects. *elba3* also shows the strongest interaction with *CP190^{P11}*, despite overall milder severity than that with *Trl*. Importantly, *ELBA*, *Trl*, or *CP190* mutant alone did not show similar defects, suggesting the interactions between ELBA and Trl or CP190 are specific.

Together, although ELBA and Insv associate with a subset of known insulator proteins, the ELBA factors seem to be selectively needed during early embryonic development in collaboration with other insulator proteins. Importantly, even though *ELBA* and *insv* mutants are viable and do not exhibit substantial embryonic patterning defects, the dose-sensitive interactions we

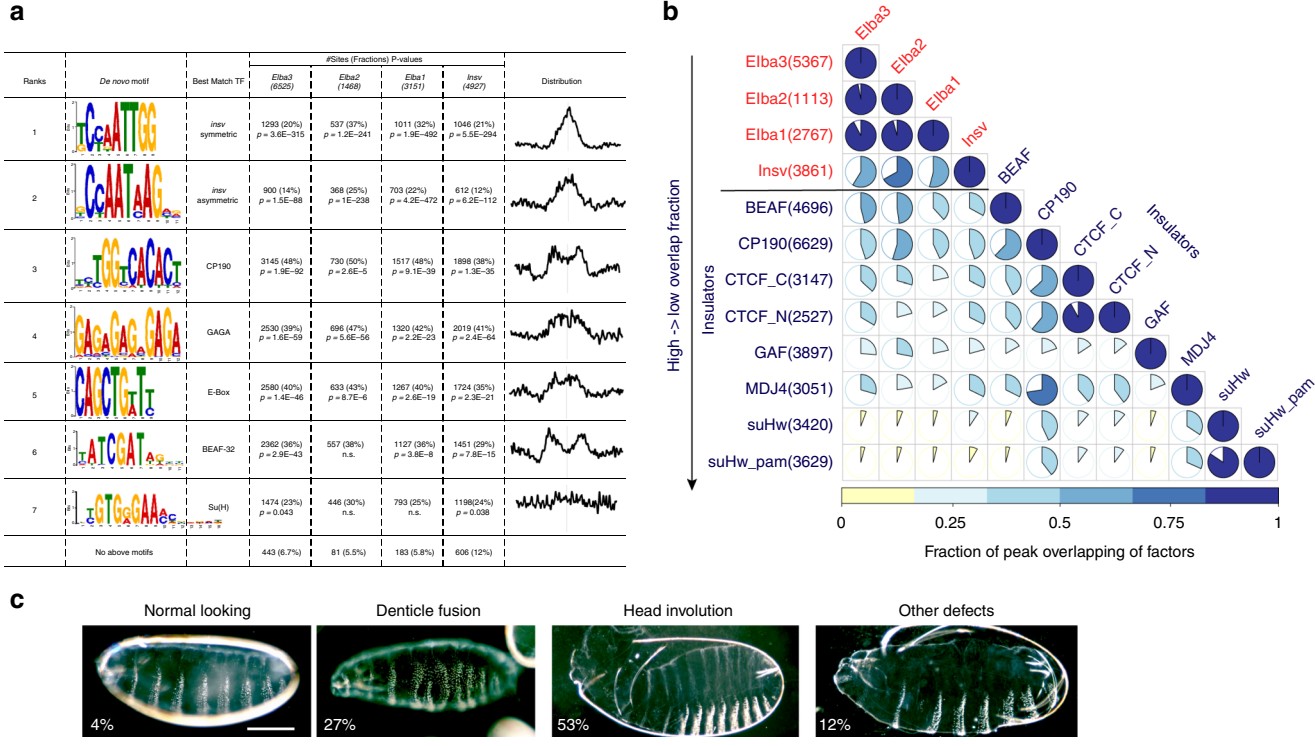

**Fig. 5 Interaction of ELBA and Insv with other insulator proteins. a** The de novo motif discovery analysis from the Elba and Insv peaks identified the Insv/ELBA symmetric and asymmetric motifs, motifs for CP190, GAF, and BEAF-32, E-box and the Su(H)-binding site. **b** The pairwise peak overlapping matrix summarizes the genomic co-occupancy of the ELBA factors, Insv and six known insulator proteins (see Methods). Among these insulator proteins, CP190 exhibits the highest overlap with ELBA and Insv, followed by BEAF-32 and CTCF. ELBA and Insv have the least overlapping with GAF, Mod(Mdg4), and Su(Hw). **c** Representative images of cuticle preps from the mutants for the *ELBA* factors and *Trl (GAF)* or *CP190*. The percentages are shown for animals with the *elba3* homozygous with the *Trl^R85* heterozygous mutations. Scale bar: 100 μm. Source data of phenotypic quantification is available in a Source Data file.

**Table 1 Genetic interactions between ELBA and Insv with Trl (GAF) or CP190.**

| With *Trl^R85* | *elba1/elba1; Trl^R85/TM3* | *elba2/elba2; Trl^R85/TM3* | *elba3/elba3; Trl^R85/TM3* | *insv/insv; Trl^R85/TM3* | *+/+; Trl^R85/TM3* |
|---|---|---|---|---|---|
| Normal looking | 66% | N.D. | 4% | 81% | 96% |
| Denticle fusion | 15% | N.D. | 27% | 1% | 3% |
| Head involution | 2% | N.D. | 53% | 0% | 1% |
| Others | 6% | N.D. | 12% | 2% | 0% |
| Total | 250 | N.D. | 143 | 127 | 311 |
| With Trl^13C | *elba1/elba1; Trl^13C/TM3* | *elba2/elba2; Trl^13C/TM3* | *elba3/elba3; Trl^13C/TM3* | *insv/insv; Trl^13C/TM3* | *+/+; Trl^13C/TM3* |
| Normal looking | N.D. | 60% | 64% | N.D. | 99% |
| Denticle fusion | N.D. | 27% | 10% | N.D. | 1% |
| Head involution | N.D. | 0% | 3% | N.D. | 0% |
| Others | N.D. | 3% | 7% | N.D. | 0% |
| Total | N.D. | 288 | 102 | N.D. | 151 |
| With CP190^P11 | *elba1/elba1; CP190^P11/TM3* | *elba2/elba2; CP190^P11/TM3* | *elba3/elba3; CP190^P11/TM3* | *insv/insv; CP190^P11/TM3* | *+/+; CP190^P11/TM3* |
| Normal looking | 72% | 96% | 58% | 100% | 100% |
| Denticle fusion | 18% | 4% | 14% | 0% | 0% |
| Head involution | 0% | 0% | 0% | 0% | 0% |
| Others | 9% | 0% | 1% | 0% | 0% |
| Total | 87 | 167 | 73 | 150 | 150 |

N.D. non determined

observe with other insulator proteins support the notion that they impact developmental gene regulation and that insulators in flies are composed of redundant elements.

**ELBA insulates adjacent transcription units.** The class I insulators are enriched in gene-dense regions and are located proximal to promoters. It was proposed that they might function to separate closely spaced transcription units[12]. As Insv and ELBA bind to class I insulators and are enriched in gene-dense regions, we asked whether these factors insulate adjacent genes to ensure transcription autonomy. As the early embryo contains abundant maternal RNAs, we used PRO-seq assay from 2–4 h wt

and mutant embryos to identify real-time transcripts produced by RNA Pol II. We then did de novo PRO-seq peak calling to define actively transcribed genes and determined differential expression between every gene pair that is separated by an ELBA or Insv ChIP peak. In the *ELBA* mutants, there is a global reduction in the differential expression of paired genes that are separated by ELBA ChIP peaks (p-values by the Bonferroni correction < 0.001, Supplementary Fig. 7a, b). We performed a Monte-Carlo simulation of expression differences between randomly chosen adjacent promoters in the genome (see the Methods section). This confirmed that the fold change between ELBA-flanked adjacent promoters is significantly higher than random. We reasoned that if the expression levels of two adjacent promoters differ more, there might be a higher need of insulation between them. This was indeed the case. For the promoter pairs that differ by more than fourfold in their expression, the reduction in differential expression became more apparent with p-values adjusted by the Bonferroni correction < 0.0001 (Fig. 6a; Supplementary Fig. 7c). In contrast, for the pairs whose expression differed by less than fourfold, no significant change was detected (Fig. 6b). All three types of promoter-pair configuration, convergent, tandem, and divergent, show a similar trend (Fig. 6). The trend of reduction is consistent when gene-body reads were used to calculate differential expression (Supplementary Fig. 7d, e). The *insv* mutant did not show such a global effect, although many individual loci display a similar reduction in the expression difference between pairs separated by Insv ChIP peaks in the *insv* mutant (Fig. 6c–e). This suggests that the insulation function of Insv-bound sites is important, but the impact may not be as global as that of ELBA-bound sites in the early embryo. Thus, we conclude that the ELBA factors globally insulate transcription units in the *Drosophila* embryo.

**ELBA-bound elements block enhancer–promoter interaction.** If ELBA/Insv binding separates unrelated promoter–enhancer interaction in the early embryo, ELBA/Insv-bound elements may be able to block enhancer interactions in ectopic settings. We tested this possibility by using an enhancer-blocking transgene which has the *LacZ*, and the *white* reporters controlled by two enhancers, *2xPE* and *iab-5* (Fig. 7a [34]). *2xPE* is an enhancer from the *twist* gene locus that drives reporter expression in a ventral stripe in the early embryos. *iab-5* is an element from the *Abd-B* region of BX-C and drives reporter expression in the posterior segments of the embryo. In between the two enhancers, there is a restriction site for inserting sequences to be tested for enhancer-blocking activity. As shown in Fig. 7a, *LacZ* and *white* will be expressed in both the *2xPE* and *iab-5* domains if the test fragment, for example, the *uMar* spacer, does not have insulation activity (Fig. 7a). We selected 11 DNA fragments that are bound by ELBA and Insv, and three control DNA sequences that are not bound (Supplementary Fig. 8, Supplementary Table. 3). Two of the three control regions showed no insulation activity. The third control transgene which contains a sequence from the *Dpr8* region gave inconsistent results between two independent lines. In contrast, six of the ELBA/Insv-bound sequences show strong blocking of the *2xPE* enhancer from activating the *lacZ* gene, and weaker blocking of the *iab-5* enhancer from activating the *white* gene (Fig. 7a; Supplementary Fig. 8, Supplementary Table. 3). Thus, many of the ELBA/Insv-bound loci behave as insulator elements in this assay.

To test whether ELBA or Insv is required for the insulation function in the reporter, we focused our analysis on the element from the *wg* gene that shows strongest blocking activity (Fig. 7a). This element contains an ELBA-type of asymmetric motif bound by the ELBA complex (Fig. 7b). In the *insv* mutant, expression of

*lacZ* remained the same as in wt embryo because the *2xPE* enhancer is still blocked (Fig. 7c). Remarkably, in the *elba3* mutant, the *lacZ* expression of the ventral stripe is recovered, suggesting that Elba3, but not Insv, is necessary for the insulation activity of this element.

The *wg* promoter is in a divergent orientation with the neighboring *Wnt4* promoter. To examine whether ELBA or Insv functions to insulate the *wg* promoter from enhancers that normally regulate *Wnt4* or vice versa, we calculated the expression ratio between *Wnt4* and *wg* in wt versus mutant conditions. The ratio of PRO-seq reads decreased substantially in all mutant conditions compared with wt (Fig. 7d), suggesting ELBA, probably also Insv, are required for separating these genes in the endogenous context.

## Discussion

Though BEN-domain containing proteins are present throughout metazoans, our knowledge of their molecular and biological functions is relatively poor. Here, we have used *Drosophila* as an in vivo model to investigate the functional properties of the fly BEN-solo proteins. We show that both ELBA and Insv repress transcription of direct target genes. However, only the ELBA factors play a role in early embryonic patterning together with other insulators. At a genome-wide level, ELBA is required for separating the transcription of differentially expressed genes.

The BEN domains of Elba1, Elba2, and Insv share similar amino acid sequences and identical protein–DNA interaction sites [21,23]. However, their DNA-binding activities seem to be complex. When expressed in cultured cells, all of these factors display high affinity to the palindromic site, while only the ELBA complex is able to bind the asymmetric site [21,23]. In vitro translated proteins of Elba1 and Elba2 can bind to both types of motifs when additional bridging factor is present [22,25]. Here, our ChIP-seq analyses confirm that in vivo Elba1 and Elba2 target the genome only through forming a heterotrimeric complex with Elba3. This suggests that the affinity of Elba1 and Elba2 binding to DNA is weak and needs to be enhanced by additional factors. The ChIP-seq analyses also demonstrate that Elba3 shows broader binding to the genome and is able to target many genomic loci in the absence of Elba1 and Elba2. The Elba3 protein does not have any known functional motif and not even a predictable DNA-binding domain. One potential factor that can bring Elba3 to chromatin is Insv. Indeed, the Elba3 peaks that are independent of Elba1/2 overlap more with Insv peaks. However, Insv is unlikely the only co-factor, as many of the Elba1/2-independent peaks do not overlap with Insv sites. Other insulator proteins with DNA-binding property, such as CP190 and GAF, are candidates that can bring Elba3 to the genome given that these factors co-occupy many genomic loci and display genetic interaction.

We used a high-resolution ChIP-nexus approach and confirm that Insv and the ELBA factors associate with both types of DNA motifs. The ChIP-nexus analyses also provided evidence that Elba1 and Elba2 in the ELBA complex bind to DNA in an asymmetric configuration. Intriguingly, at some of the loci, such as the asymmetric sites in *Fab-7* and *Parp*, Elba1 and Elba2 show "+" versus "−" strand preference. The genomic loci with asymmetric binding are expected to represent weak association of ELBA with DNA, as strong DNA binding would allow equal pull down of the subunits with the antibody against any of the three components. There are many other loci showing symmetric read distribution for Elba1 and Elba2. These sites either mediate strong binding of the complex or symmetric binding of Elba1 and Elba2 (e.g., as homodimers). Insv binding is always symmetric, suggesting that it binds to the sites as homodimers.

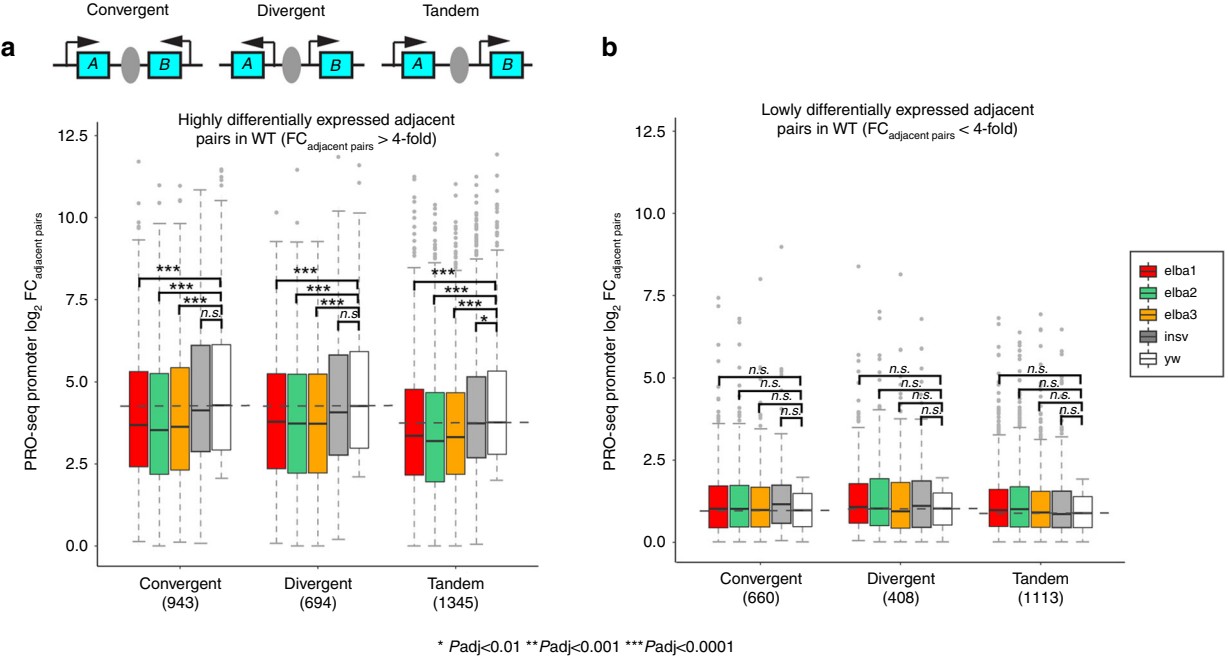

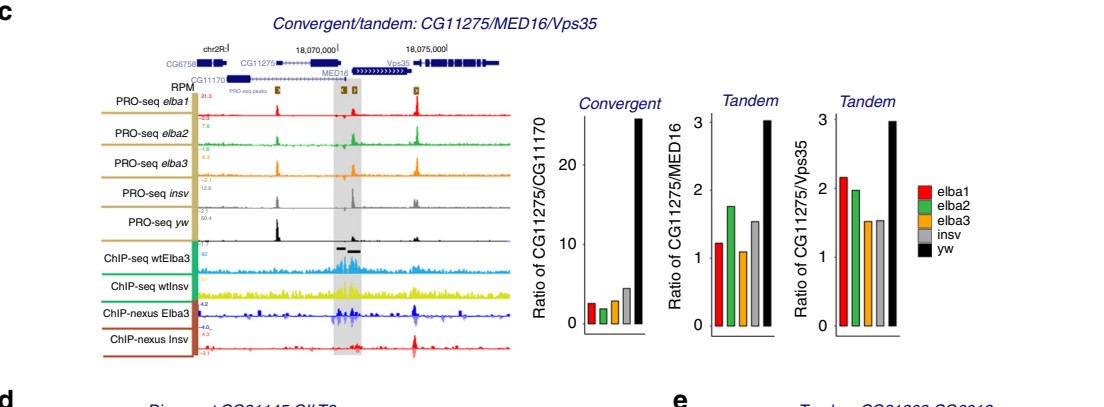

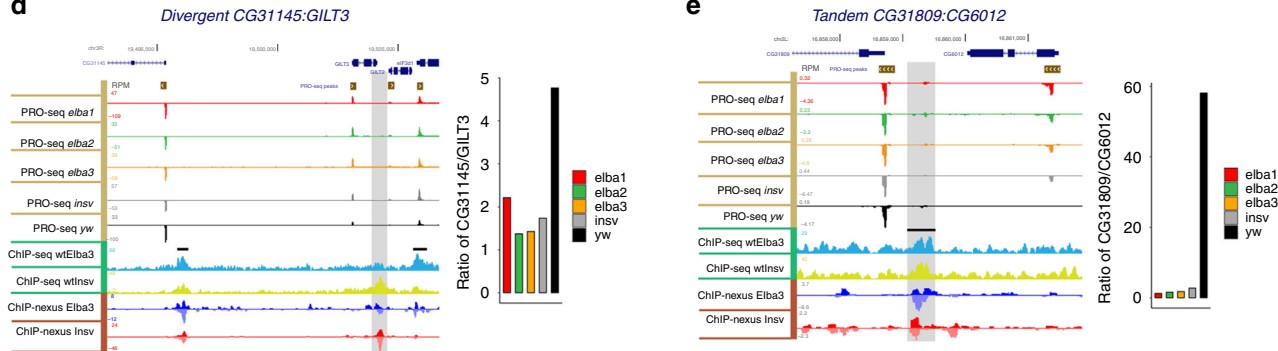

**Fig. 6 The Elba factors insulate adjacent transcription units.** The PRO-seq data from wt and mutant embryos were used to identify real-time transcripts produced by RNA Pol II in each genotype, and de novo PRO-seq peak calling was made to define active promoters. **a, b** We considered three types of adjacent promoter-pair configurations, convergent, divergent, and tandem, flanking an ELBA/Insv peak. Differential expression between adjacent promoter pairs (absolute FC $_{adjacent\ pairs}$) were divided into (**a**) the highly differentially expressed (>fourfold) and (**b**) the lowly differentially expressed (< fourfold) pairs in the wild type (wt). **a** Significant reduction of expression difference between highly differentially expressed neighbor promoters in the three *ELBA* mutants, but not in *insv* mutant. **b** Lowly differentially expressed gene pairs do not show significant changes in any of the mutants. Statistical significance was calculated using two-tailed *t* tests, and the *p*-values were adjusted by the Bonferroni multiple testing correction method (*$p < 0.01$, **$p < 0.001$, ***$p < 0.0001$). **c–e** Three exemplary loci with convergent, divergent, or tandem gene pairs flanked by the ELBA/Insv-binding peaks. The bar plots on the right show the ratio of expression (RPM) of the two adjacent promoters in each genotype determined by PRO-seq read coverage of the promoter-proximal regions. The boxplots are defined: center line, median; box limits, upper and lower quartiles; whiskers, 1.5x interquartile range; points, outliers.

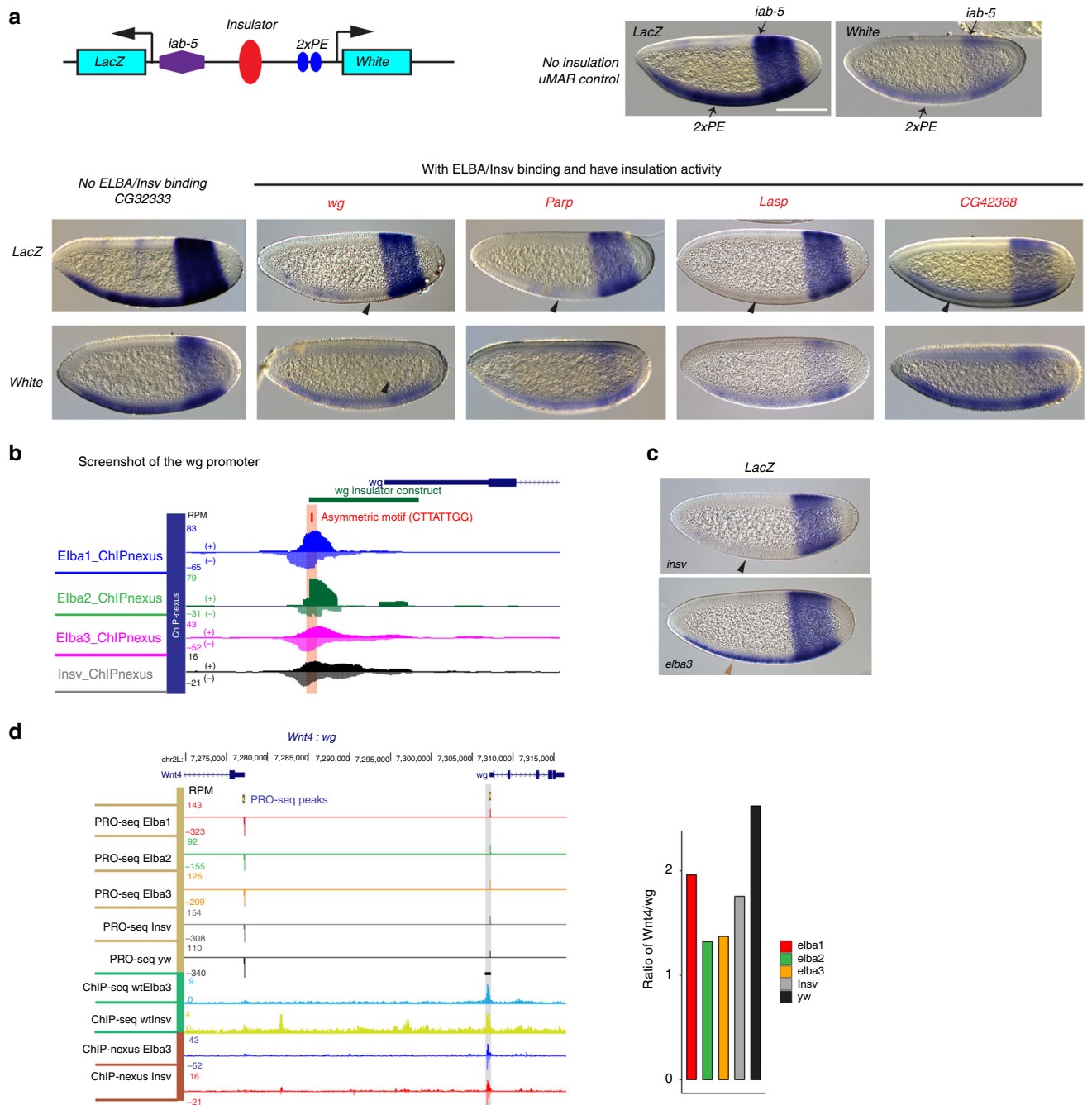

**Fig. 7 ELBA is required for separating transcription of *Wnt4* and *wg*. a** Transgenic insulator assay. In situ hybridization images show expression of the *lacZ* and *white* genes driven by the *2xPE* and *iab-5* enhancers in the ventral and the posterior stripes, respectively. An inserted control fragment (*uMAR*) does not affect the reporter expression, neither does the *CG32333* fragment that has no Elba or Insv binding. The fragments from the *wg*, *Parp*, and *Lasp* loci, highlighted in red, show blocking activity, evidenced by loss of the ventral stripe expression of *LacZ*. The *iab-5* enhancer was less affected, with a weaker but visible posterior stripe of *white*. Black arrowheads indicate the weakened or lost staining by insulation activity of the inserted fragments. **b** A screenshot of the insulator in the *wg* locus with ELBA and Insv binding and an asymmetric Insv/ELBA motif. **c** The *lacZ* staining shows loss of insulation activity of the *wg* element in *elba3*, but not *insv* mutant embryo. The black arrowhead indicates the absence of the ventral stripe. The brown arrowhead indicates recovery of the ventral stripe due to loss of insulation in the *elba3* mutant. **d** A screenshot of the divergent pair *Wnt4* and *wg* with an Elba/Insv peak proximal to the *wg* promoter. The ratio of PRO-seq promoter expression of *Wnt4* versus *wg* decreased in the mutants compared with wt. Scale bar: 100 μm.

This evidence shows ChIP-nexus can be a powerful tool to resolve binding symmetry by a heterotrimeric complex. It will be of interest to understand how the BEN domains have evolved in DNA-binding affinity and sequence specificity across species. Our previous and current work well exemplifies the approach to determine the molecular properties of a less studied DNA-binding protein family[21,23].

The activity of ELBA in the early embryo was examined by RNAi knockdown experiments, where it was shown to influence early boundary activity of the HS1 element[22]. However, the effect of complete loss of ELBA in embryonic development has not been investigated. We generated the loss-of-function mutants for the ELBA genes and found that they are dispensable for viability. This is not surprising as other chromatin insulator proteins, such as

dCTCF[35] and BEAF-32[36], are not required for viability. One possibility is that *Drosophila* utilizes multiple backup mechanisms to ensure boundary fidelity. Indeed, when one copy of *CP190* or *GAF* is removed, loss of ELBA led to drastic developmental consequences in this sensitized background. Despite both ELBA and Insv associating with known insulator proteins, such as CP190, BEAF-32, and GAF (ref. [23]; Fig. [5]), *ELBA* showed strong genetic interactions with CP190 and GAF in viability and early embryonic patterning, while *insv* did not. It is possible that Insv is less needed during the embryonic stage and/or that another unknown factor compensates for its joint function with CP190 and GAF. In support of the first possibility, *insv* is required for maintaining segmentation of adult flies when the GAF sites are mutated from *Fab-7*[25].

Genes in the *Drosophila* genome are more compact than in vertebrates. There is a need to partition dense transcription units to ensure enhancer specificity. Thanks to many years of genetic studies in *Drosophila*, a list of individual genomic loci were identified that separate enhancers or promoters[37–42]. Insulator proteins such as GAF, CTCF, CP190, and BEAF-32 were found to mediate these activities. It was shown that BEAF-32 binds to sequences in between closely apposed genes with a head-to-head configuration (divergent)[43]. We show that the neighboring genes that are differentially expressed in wild type become more equally expressed in *ELBA* mutant embryos. In this case, all three types of promoter configurations, divergent, tandem, and convergent, have similar requirement for ELBA-dependent insulation. Together with the evidence that genomic elements bound by ELBA and Insv are sufficient to block enhancer–promoter interactions in transgene assays, our results suggest ELBA is required to ensure the autonomous regulation of linked transcription units.

New properties have been assigned to insulators recently, especially in chromatin organization and long-range cis-element interactions. In this work, we focus on the functions of ELBA and Insv in active chromatin regions because of their enrichment in close proximity to active promoters. However, we have detected enrichment of ELBA and Insv in several known elements that could mediate long-range interactions, such as the *homie-nhomie*[44] and *scs* and *scs'* loci[6,45]. Future studies will be needed to determine the roles of ELBA and Insv in chromatin organization.

## Methods

**Fly strain culturing and generation of transgenes.** All fly stocks were kept at 25 °C. The insv mutant allele insv[23B] was described previously[46]. *ELBA* mutants were created using CRISPR: transgenic flies carrying single-guide RNA targeting the coding sequence of each gene were crossed into the nos-Cas9 transgenic flies. Frame-shift mutations were identified by PCR and Sanger sequencing. The *Trl*[R85] and *Trl*[13C] alleles were kindly provided by Dr. Ana Busturia (Centro de Biología Molecular "Severo Ochoa" CSIC-UAM), the *CP190*[P11] allele was from Bloomington Stock Center and used for genetic interaction crosses.

For making the insulator transgenes, selected fragments were amplified and cloned into the insulator transgene backbone (kindly provided by Dr. Jumin Zhou[34]). The sequences of cloning oligos are provided in Supplementary table 7. All transgenic flies were created at BestGene, Inc.

**Cuticle preparation.** Embryos were collected and aged to 24–36 h before dechorionation with bleach. They were rinsed, directly mounted in 85% lactic acid, and cleared at 60 °C for 3–6 h.

**Immunostaining.** Embryos were collected, dechorionated with bleach, and fixed with 4% formaldehyde in PEM (0.1 M PIPES, 2 mM EGTA, 1 mM MgSO4, pH 7) for 20 min. For immunostaining, embryos were rehydrated, washed with PBTween (0.1% Tween in PBS), and blocked with PBSBT (0.1% Triton X-100, 1% BSA in PBS). The primary rabbit anti-Elba1 or anti-Elba3 were added to the samples in 1:150 and 1:500 dilution with PBSBT, respectively, and incubated at 4 °C overnight. After extensive washing with PBSBT (5–15 min), the secondary antibody (Alexa488 goat anti-rabbit, Jackson ImmunoResearch) was added in 1:500 dilution and incubated at room temperature for 2 h. After washing with PBSBT and then

PBTween, the embryos were counterstained with DAPI and mounted in Vectashield (Vector Laboratories).

**In situ hybridization.** The *LacZ* and *white* probes were generated by transcription from linearized pBluescript template plasmids (kindly provided by Dr. Mattias Mannervik) with T3 or T7 RNA polymerase (Thermo Fisher) and Dig RNA labelling mix (Roche) according to the manufacturer. Embryos were aged and fixed with 9% formaldehyde in fixation buffer (1.3x PBS, 67 mM EGTA, pH 8) for 25 min. For in situ hybridization, fixed embryos were permeabilized with xylene and rehydrated as well as postfixed with 5% formaldehyde in PBT (1x PBS, 0.1% Tween-20) for 25 min. Embryos were treated with proteinase K (4 μg/ml) for 8 min, followed by another round of postfixation for 25 min, before hybridization with the probes at 55 °C for overnight in hybridization buffer (50% formamide, 5x SSC, 100 μg/ml sonicated boiled ssDNA, 0.1% Tween-20). Samples were incubated with alkaline-phosphatase-labeled anti-digoxigenin antibody (1:2000, Roche) overnight at 4 °C, and developed with 0.6 mg/ml nitrotetrazolium blue chloride (NBC) and 0.3 mg/ml 5-bromo-4-chloro-3-indolyl phosphate disodium salt (BCIP). Samples were dehydrated by repeated washes in ethanol, rinsed in xylene, and mounted in Permount (Fisher).

**Co-immunoprecipitation (Co-IP) and western blot.** Embryos from wild-type (wt), *elba1*[SK6], *elba2*[SK2], *elba3*[SK3], and *insv*[23B] were collected and aged for 2–4 h. In each IP, 50 μl of embryos were used. Protein extracts were prepared by homogenizing the embryos in lysis buffer (20 mM Tris pH 8.0, 150 mM NaCl, 10 mM MgCl2, 2 mM EDTA, 10% glycerol, 0.5% NP-40, 1 mM DTT, 1x protease inhibitor cocktail [Roche]). About 5 μg of rabbit anti-Elba1, anti-Elba3, and rabbit IgG were used in the first round of immunoprecipitation. The protein/antibody complex was then precipitated with the Gamma-bind G beads (Thermo Fisher) and separated on 10% of SDS-PAGE for western blotting. The left supernatant was re-precipitated with the antibodies against Elba1, Elba3, and Insv[46] (1 μg of antibody/IP, the second round of IP). The same antibodies were used in blotting. To minimize unspecific signal from IgG heavy chains, light-chain-specific secondary antibodies of mouse anti-rabbit IgG (1:40,000 Jackson ImmunoResearch) were used, and the signals were developed with ECL Plus reagent (GE Healthcare).

**Cell culture and luciferase assay.** To generate the TetR-DBD fusions with the ELBA factors and Insv, the open-reading frames of Elba1, Elba2, Elba3, and Insv were PCR amplified and cloned into the pAC-TetR vector. All transfections were performed using *Drosophila* S2-R + cells grown in Schneider *Drosophila* medium containing 10% fetal calf serum. Cells were co-transfected with TetR fusion, 2xTetO-Firefly luciferase and pAc-Renilla plasmids in 96-well plate using the Effectene Transfection kit (Qiagene). Luciferase assays were performed and measured using the Dual Luciferase Assay System (Promega). Expression was calculated as the ratio between the firefly and *Renilla* luciferase activities. Three rounds of transfections were performed on different days and different cell populations, and considered as three biological replicates. Within each biological replicate, four wells of cells were transfected with the same DNA mixture and measured and thus considered as four technique replicates. In total, for each DNA combination, the values from the 12 wells of transfection were calculated, averaged, and plotted in Fig. [4a]. Cloning oligos are provided in Supplementary Data 6.

**ChIP-qPCR and ChIP-seq.** For chromatin immunoprecipitation (ChIP), embryos from wild-type (wt), *elba1*[SK6], *elba2*[SK2], *elba3*[SK3], and *insv*[23B] were collected and aged for 2–4 h (50 μl of embryos per reaction). After dechorionation in bleach and rinsing with water, embryos were homogenized in 1.8% formaldehyde and 2.5 mM DSG (Di(N-succinimidyl) glutarate, Sigma) in buffer A1 (60 mM KCl, 15 mM NaCl, 15 mM HEPES pH 7.6, 4 mM MgCl2, 0.5 mM DTT, 0.5% Triton X-100, and 1x inhibitor cocktail (Roche)). Chromatin was extracted by repeatedly spinning at 4000xg and washing in buffer A1 (3x total). Cross-linking occurred for 10 min on ice in LysisPlus buffer (140 mM NaCl, 15 mM HEPES pH 7.6, 1 mM EDTA, 0.5 mM EGTA, 0.1% sodium deoxycholate, 1% Triton X-100, 0.5 mM DTT, 0.1% SDS, 0.5% sarcosyl and 1x inhibitor cocktail (Roche)). Sonication was done using a Bioruptor (Diagenode) in high-power mode, 15 cycles of 30 s ON/30 s OFF. The samples were pre-cleared with Gamma-bind G sepharose beads (GE healthcare) overnight and the beads removed. The ELBA antisera used for IP were tested in ChIP previously[26] and kindly provided by Dr. Paul Schedl (Princeton University). Two different sets of antibodies were used for ChIP-qPCR and ChIP-seq, respectively. For each ChIP reaction, 5 μl of antibody were added to the respective sample, and the tubes rotated at 4 °C overnight. New beads were then added to the samples and binding occurred for 4–16 h before transferring the beads in 200 μl LysisPlus buffer (without sarcosyl) onto Ultrafree Filter columns (0.45 μm, Millipore). The beads were washed on the filters four times for 10 min with LysisPlus (no sarcosyl) and twice with TE (10 mM Tris-HCl pH 8.0, 1 mM EDTA). Elution of the samples from the beads occurred by first adding 100 μl EB1 (10 mM EDTA, 1% SDS, 50 mM Tris-Cl pH 8) for 30 min at 65 °C, then adding 100 μl EB2 (TE + 0.67% SDS) for 30 min at 65 °C, and spinning the whole elute through the filters at 3000 rpm for 2 min in a tabletop centrifuge. All samples as well as input samples (in 200 μl TE + 1%SDS) were reverse cross-linked at 65 °C overnight, digested with 2 μl 20 mg/ml proteinase K (Thermo Fisher) for 2 h at 42 °C, and RNase digested

with 2 μl 0.5 mg/ml RNase (DNase-free, Roche). The DNA was purified using the QIAquick PCR Purification Kit (Qiagen) with two times elution in 30 μl each.

All qPCR reactions were run in technical triplicates using iTaq Universal SYBR Green Supermix (Bio-Rad). In all, 1 μl of chromatin template was used in each 20-μl reaction, and amplification was normalized to 0.2% input. qPCR oligos are provided in Supplementary Data 6.

ChIP-seq libraries were made using the NEBNext Ultra™ II DNA Library Prep Kit and sequenced in the Illumina Hi-seq platform. The ChIP-seq reads were mapped to the *Drosophila melanogaster* (dm3) genome assembly using Bowtie2 with the default parameters, after the adaptor trimming by Trimmomatic. The uniquely mapped reads with a mapping quality MAPQ > 20 were used for further analysis. For all ChIP-seq samples, we generated coverage tracks at 1-nt resolution and normalized to the library sizes to give read per million (RPM) in "bigwig" format. We further created the coverage differential tracks for four factors by subtracting the mutant from wild-type coverage (log2 wt/mutant).

For each of the four factors, the peak calling was performed by the ChIP-seq reads of wt or a mutant condition to its own mutant ChIP or IgG or Input. The peaks were called using MACS2[47] with default parameters and the confident peaks were determined by an FDR < 1%. The peaks overlapped with *Drosophila* blacklist were also removed. Peak overlap analysis was performed by "mergePeaks" function in Homer2 package with the default parameters, and a maximum distance was set to "-d given" which requires the peak regions overlap.

The de novo motif search was performed for all the called peaks for each factor by MEME-ChIP[48]. We extended the summits of the called peaks by 500 nucleotides in each direction, and searched for 5–15 nt motifs in the central regions (100 nucleotides) using default parameters.

Pairwise peak overlaps were performed for the ChIP-seq peaks of Elba and Insv factors with the modEncode insulator data sets (Negre et al.[12]), including ChIP–ChIP data for CP190, BEAF-32, CTCT, GAF, Mod(Mdg4), and Su(Hw). As ChIP-seq peaks are generally narrower than ChIP–ChIP peaks, we extended the ChIP–ChIP summits by 100 nucleotides in each direction for the overlapping analysis. We used a maximum distance of 50 nucleotides between the peak summits for overlapping. The overlap fraction between the two sets was calculated by the number of overlapped peaks divided by the minimum number of peaks of two sets.

**ChIP-nexus analysis.** ChIP-nexus was performed following the step by step protocol described in He et al.[28]. In total, 20 μl of each antibody and 200 μl of embryos were used in each ChIP-nexus reaction. All the ChIP-nexus libraries were sequenced on the Illumina Hiseq2500 platform with 1 × 50 bp SR configuration.

Before aligning the ChIP-nexus reads to the genome, the 5′ fixed barcode (1–5) was first removed, and the random 4nt barcode was retained for each read. After the 3′ adaptor trimming by Trimmomatic, the sequencing reads were collapsed to only include unique reads. The random 4nt barcode was further removed, and the reads with at least 22 nucleotides were retained for mapping. We mapped the reads using bowtie with the parameter setting "-k 1 -m 1 -v 2 --best --strata". Similar to ChIP-seq, we generated normalized coverage tracks separately for each strand in "bigwig" format. Similar to the ChIP-seq data, the ChIP-nexus peak calling was performed by MACS2 using the default parameter. To obtain highly confident binding sites for each factor, we set a highly stringent cutoff (FDR < 1E-10 for Elba1, Elba3, and Insv, and FDR < 1E-5 for Elba2). We further required the binding sites to be called by both ChIP-seq and ChIP-nexus. Defined high-confident peaks are summarized in Supplementary Data 2.

To examine the asymmetry of the binding sites, we calculated an OI for each binding site by ChIP-nexus for each factor. OI was defined by maximum number of reads between two strands divided by the sum of reads of two strands, max (forward, reverse)/sum (forward, reverse), ranging from 0.5 to 1. The OI values for the peaks are included in Supplementary Data 3.

**RT-qPCR, RNA-seq, and analysis.** The total RNA was extracted from stages 2–4 h embryos using Trizol reagent (Invitrogen). RNA quality was tested by the Agilent Bioanalyzer.

For RT-qPCR, the total RNA was first cleared from residual DNA using Turbo DNA-free kit (Invitrogen), and were then reverse-transcribed using Superscript III Supermix (Invitrogen). qPCR primers for Elba1, Elba2, and Elba3 were so designed with one primer binding on top of the CRISPR cut site and those for insv are inside of the deletion of the insv23B allele. The qPCR reactions were run in technical triplicates using iTaq Universal SYBR Green Supermix (Bio-Rad). Amplification for each sample was normalized to the housekeeping gene RpL32. Four biological replicates were performed. qPCR oligos are provided in Supplementary Data 6.

RNA-seq libraries were made using the Illumina Truseq Total RNA library Prep Kit LT. Sequencing was performed on the Illumina Hiseq2500 platform.

After trimming the adaptor sequences using Trimmomatic, the RNA-seq reads from the replicated wild type (x3), and mutant samples (x3) were mapped to the *Drosophila melanogaster* (dm3) genome assembly using HISAT2. RNA-seq signal was normalized by the TMM method implemented in the Limma Bioconductor package[49]. The gene annotation was obtained from FlyBase. Differentially expressed mRNAs between BEN factors mutants versus wild type were identified, and FDR (Benjamini–Hochberg) was estimated. Calculated differential expression is presented in Supplementary Data 4.

To test whether a set of genes are significantly changed (up- or downregulated as a gene set) among the differentially expressed (DE) genes from wild-type and mutant RNA-seq data, the gene set enrichment testing function "camera" in the R limma package was used. "camera" is a ranking based gene set test accounting for inter-gene correlation, to test whether the called peaks in five Elba3-binding subsets are significantly changed as a set.

**PRO-seq assay and analysis.** The protocol was adapted from Kwak et al.[50]. Drosophila embryos of 3.5–4.5 h age were dechorionated and homogenized in 1 ml of buffer A (10 mM Tris-Cl pH 8.0, 300 mM sucrose, 3 mM CaCl₂, 2 mM MgAc₂, 0.1% Triton X-100, 0.5 mM DTT). The homogenate was filtered through a nylon membrane (pore size 0.8 μm) and spun down at 500×g. The pellet was resuspended in fresh buffer A and spun down several times for washing. Nuclei were then resuspended in buffer D (10 mM Tris-Cl pH 8.0, 25% glycerol, 5 mM MgAc₂, 0.1 mM EDTA, 5 mM DTT) to achieve a final concentration of roughly 106 cells per 5 μl before freezing in liquid nitrogen. Run-on reaction was performed for 3 min at 30 °C after adding 2x reaction mix to the samples (10 mM Tris-Cl pH 8.0, 5 mM MgCl₂, 1 mM DTT, 300 mM KCl, 0.05 mM of each of the four Biotin-NTP, 0.4 U/μl RNAse inhibitor, 1% sarkosyl). Nascent RNA was extracted directly after with Trizol, precipitated in ethanol, and fragmented by base hydrolysis in 0.2 N NaOH on ice for 10 min. After neutralization with 1 M Tris-Cl pH 6.8, salts and free NTPs were removed by buffer exchange on a P-30 column (Bio-Rad). To enrich biotin-labeled RNA, the samples were bound to 30 μl M280 streptavidin beads (Invitrogen) according to the manufacturer and washed with twice with high salt buffer (2 M NaCl, 50 mM Tris-Cl pH 7.4, 0.5% Triton X-100), twice with binding buffer (300 mM NaCl, 10 mM Tris-Cl pH 7.4, 0.1% Triton X-100) and once with low salt buffer (5 mM Tris-Cl pH 7.4, 0.1% Triton X-100). RNA was extracted from the beads twice with 300 μl Trizol in each round and precipitated with glycoblue in ethanol. RNA pellets were redissolved in 4 μl of 12.5 μM reverse 3′ adaptor dilution (Rev3, see Supplementary Table 9) and ligated overnight at 4 °C with T4 RNA ligase (NEB) in 10 μl reaction volume (1x T4 RNA ligase buffer, 1 mM ATP, 10% PEG, 1U/μl RNAse inhibitor). The bead binding and Trizol extraction was repeated as above and 5′-decapping performed using RppH (NEB) by dissolving pellets in 10 μl H₂O, adding 40 μl of the reaction mix (1x NEBuffer 2, 2 U/μl RppH, 2 U/μl RNAse inhibitor) and incubating at 37 °C for 1 h. 5′-hydroxyl was performed by directly adding PNK (NEB) reaction mix to each decapped sample. The total reactions of 100 μl volume (1x PNK buffer, 1 mM ATP, 2.5 U/μl PNK, 0.2 U/μl RNAse inhibitor) were incubated at 37 °C for 1 h. The RNA was then extracted using Trizol and precipitating in ethanol. Pellets were redissolved in 4 μl of 12.5 μM reverse 5′ adaptor (VRA5, see Supplementary Table 9) and ligation performed as with the 3′ adaptor. The RNA was enriched again by bead binding and extraction as above and resuspended in 10 μl H₂O. Reverse transcription was performed using SuperScript III (Invitrogen) in 20 μl of final reaction volume. Water, dNTPs (final 0.5 mM) and reverse transcription primer (RP1 TrS, see Supplementary Table, final 2.5 μM) were added first, samples heated to 70 °C for 2 min, chilled on ice for 2 min and the remaining reaction components except for the enzyme added (1x first-strand buffer, 5 mM DTT, 1 U/μl RNAse inhibitor). After incubation at 37 °C for 5 min, the enzyme was added (15 U/μl) and the reaction run in a PCR machine: 45 °C for 15 min, 50 °C for 40 min, 55 °C for 10 min, 70 °C for 15 min. The ideal amount of amplification cycles was determined by makig serial dilutions of the template and test amplification with primer RP1 TrS, primer RPI TrS short (see Supplementary Table 9) and Phusion polymerase 2x master mix (Thermo Fisher). PCR cycling: 95 °C – 2 min, 5 × [95 °C – 30 s, 56 °C – 30 s, 72 °C – 30 s], 9 × [95 °C – 30 s, 65 °C – 30 s, 72 °C – 30 s], 72 °C – 10 min. Each library dilution row was run on an 8% polyacrylamide gel, and the ideal cycle number determined by choosing the amplification that produced a visible but not overamplified 125–350 bp fragment library. Full-scale amplification was then performed using barcoded reverse primers (RPIxx, see Supplementary Table 9). The amplified libraries were purified using 1.8x volume ratio AMPure XP beads (Agencourt). The libraries were then separated on a 0.5× TBE 8% polyacrylamide gel and cut from around 125 bp (just above visible primer-dimer) to 350 bp. The samples were extracted from the gel fragments by shredding the fragments and incubating them in twice the amount of extraction buffer (10 mM Tris-Cl pH 8, 0.5 mM NaAc, 10 mM MgAc₂, 1 mM EDTA, 0.1% SDS) at 50 °C for 13 h. After spinning down at 10,000×g for 5 min and retrieving the eluted library, a second extraction round was performed with 600 μl buffer for 2 h and the supernatants pooled. Eluates and remaining gel fragments were filtered through Spin-X filters (Corning), and the volume reduced to 500 μl in a speed vacuum centrifuge. The DNA was then extracted using buffered phenol–chloroform, and the concentration measured. Barcoded libraries were pooled and sequenced on the Illumina Hiseq2500 platform with 1 × 50 bp SR configuration.

The adaptors were first trimmed from the sequencing reads by cutadapt software and the reads with at least 15 nt were retained. We then removed reads that mapped to rRNAs, and the remaining reads were further mapped to the *Drosophila melanogaster* (dm3) genome assembly using BWA with the default parameters. We also generated the PRO-seq coverage tracks (normalized by the library sizes) with separate strands for each factor. To detect de novo transcripts from PRO-seq, we combined all genotypes and adapted the Homer2[51] GRO-seq transcript identification method (using a parameter setting "findPeaks -style groseq -tssFold 4 -bodyFold 3"). The pausing regions (promoter region) were defined from

the de novo transcript starts to 200 nt downstream, and the gene-body regions were defined from 400 nt downstream to the end of the de novo transcripts. The de novo transcripts having a promoter expression of greater than 1 transcript per million (TPM) were retained for further analysis. Fold difference between adjacent gene pairs is summarized in Supplementary Data 5.

**Analysis of ELBA/Insv factors acting as insulators**. For each Elba/Insv high-confident binding site, which was called by both ChIP-seq and ChIP-nexus (see Methods above), we looked for the adjacent PRO-seq promoter pairs. We then calculated absolute differential expression between the adjacent promoter pairs (abs log2FC adjacent pair). We further classified the adjacent promoter pairs flanking an Elba/Insv peak into three types: convergent, divergent, and tandem.

To test whether the expression change between the adjacent pairs is above background, we performed a Monte-Carlo simulation. We randomly located the same number of regions with the same length as the Elba/Insv ChIP peaks in the same chromosome and repeated the random selection and calculation 2000 times. *P*-values were calculated by dividing the number of instances that show a higher fold change between the random adjacent genes than that between the Elba/Insv-bound genes by 2000 iterations. These were done separately for convergent, divergent, and tandem pairs in each of wt and four mutants.

## Data availability
The data supporting this study are available from the corresponding author(s) upon reasonable request. The sequencing data are deposited in GEO (GSE131160). Library statistics are included in Supplementary Data 1. Raw qPCR values and uncropped blots are available in a Source Data file.

## Code availability
The R code for the data analyses and generating figures are available at Github: https://github.com/jiayuwen/ELBA.

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

## Acknowledgements
We thank the imaging facility (IFSU) of Stockholm University and the National Genomic Institute of Scilife Laboratories, Sweden, for providing service and support. The Bloomington Stock Center and the *Drosophila* community provided important reagents and fly stocks. We thank Brian Joseph for preliminary analysis of ChIP and expression data. J.W. was supported by the Australian Research Council (ARC) Future Fellowship (FT160100143) and ANU Futures Scheme. Work at P.S. laboratory was supported by the grant GM R35GM126975. Work in E.C.L.'s group was supported by the National Institutes of Health grants R01-NS074037 and R01-NS083833 and Memorial Sloan Kettering core grant P30-CA008748. The project was supported by the Young Investigator grant from Swedish Research Council (Vetenskapsrådet, 2014-5584) to Q.D.

## Author contributions
Q.D. conceived and designed the project. E.L. helped the initiation of the project. M.U. and Q.D. performed most of the experiments. H.W. performed the luciferase and co-immunoprecipitation experiments. C.Z. generated the constructs for luciferase assays. S.K. generated the *elba* mutants. P.S. and T.A. provided critical information on the Elba antibodies. J.W. performed all the computational analysis. Q.D. and J.W. analyzed and interpreted the data with input from M.U. and H.W. Q.D. wrote the paper with help from J.W. and input from other co-authors.

## Competing interests
The authors declare no competing interests.
