## [Peer Review File · Nature Communications]

Reviewers' comments:

Reviewer #1 (Remarks to the Author):

The manuscript submitted by Ueberschär et al. presents a genome-wide analysis of both the binding of *Drosophila* BEN-solo factors (+Elba3) and an analysis of the transcriptional repercussion of their individual mutations in an *in vivo* context. The authors dissect the binding interplay of the different ELBA sub-units along with Insv and show that both Elba3 and Insv have a substantial number of independent binding sites. Furthermore, they confirm their role in gene repression and go on to show a role in insulation between neighboring genes.

Chromatin insulation is a very interesting subject, and here, the authors provide a considerable amount of genome-wide data (ChIP-seq, RNA-seq, PRO-seq) that will most likely be of interest to the *Drosophila* community. Additionally, the last part of the article, focusing on the insulating capacity and its developmental impact, while not particularly novel, is quite convincing. However, I must admit that I find the first half of the manuscript difficult to follow and several claims reported throughout the article are overstated. While a considerable amount of work has gone into this manuscript, unfortunately, in its current state, I cannot recommend it for publication in *Nature Communications*.

Here are some points that I would recommend addressing before further consideration.

Major points:

- The first half of the results section is dedicated to ELBA and Insv genome-wide binding characteristics. While this data, in a more succinct form, would be better suited for publication, what I fail to grasp is why the authors spend so much time analyzing ChIP-seq data which is clearly inferior in quality to their ChIP-Nexus results. Given the non-normalized ChIP-seq profiles compared to the ChIP-Nexus profiles (Fig. 3A), I don't see how the ChIP-seq data could prove more trustworthy and would recommend using the ChIP-Nexus data as a primary source and the ChIP-seq as a backup and not vice-versa. Furthermore, if ChIP-Nexus is supposed to provide "better specificity and resolution" as the authors claim, why are there so many less peaks called than in ChIP-seq (Fig. 1B vs Supp. Fig. 3B)? Does this mean that a large proportion of ChIP-seq peaks are false-positives?
- Furthermore, in several instances, the authors mention their Elba2 ChIP-seq. While upon normalizing to mutant ChIP-seq they manage to call certain peaks (Fig. 1B), Supp. Fig. 1B clearly shows Elba2 enrichment to be suboptimal for further analysis.
- One of the major findings of the paper is that both Elba3 and Insv have a large number of binding sites to which they are recruited independently of the other BEN-solo factors. Yet, the authors fail to delve into the description of the particularities of these sites beyond genetic localization and motif representation (Supp. Fig. 3F). It would have been interesting to know if different categories of genes are regulated by solo rather than by complex binding. Or do solo sites have a different effect on gene regulation than complex sites (according to Supp. Fig. 4, independent Insv sites seem to upregulate approximately as many genes as downregulate them). Or yet, do these proteins co-localize with different sets of proteins depending on if they bind solo or in a complex?
- The same case could be made about the focus on binding sites containing motifs. While I understand that the motifs offer some confirmation as to the veracity of these binding sites, the fact that 71 and 75% for of Elba3 and Insv binding sites (Supp. Fig. 3A), respectively, show no typical motif enrichment is both puzzling and fascinating. Yet, no mention is made of this.
- A certain number of claims are made by the authors to which I find the data to prove them insufficient. Examples:
 - in elba2 mutants, Elb1, Elba3 and Insv are unlikely to form a complex.
 - Insv is involved in recruiting Elba3 (especially since in the next paragraph, the authors study

Elba3 recruitment in a cell-type that does not express Insv).

- ELBA and Insv only repress transcription and the down-regulated genes are due to "random overlapping" (this is particularly hard to assess since there is no further analysis of these genes/binding sites).
- The insulation function of Insv-bound sites is restricted to "certain pairs of genes" ◊ too vague.

Minor points:

- The manuscript contains a large number of typos and grammatical errors which sometimes make it hard to follow (such as Mdg4/Mdj4, cortex/contexts, etc...).
- I'm not quite clear on how Supp. Fig. 1A shows that the "peak expression" levels of the ELBA factors are during the blastoderm stage of the embryos.
- While the immunofluorescence images in Supp. Fig. 1A are appreciable, they do not allow for precise quantification of the effects of mutating the Ben-solo + Elba3 genes on each other. Therefore, a quantified Western-blot, or, if no antibodies of sufficient quality are available, qRT-PCR experiments of all analyzed factors in all mutant conditions should be presented in order to truly appreciate the effect of the mutations. This would also allow to show the effects of Elba2 mutation.
- If my understanding is correct, Supp. Fig. 1C shows the genetic positioning of Elba3 with and without Insv. However, in this figure when adding both amounts (3729+2555) we fail to reach the 6525 called peaks mentioned in the text. The same is true for Insv.
- Fig. 1D, Elba2 data is missing.
- While I don't disagree with the statement, I believe simple tag-density plots showing Elba3 ChIP-seq signal enrichment in the different types of scenarios (with or without Elba1/2 or Insv) would be easier to understand and perhaps more informative than the cumulative distributions of peak scores (ie Supp. Fig. 2E).
- Similarly, rather than showing a browser shot (with a tiny y-axis comparted to other browser shots, possibly indicating background noise), Supp. Fig. 3E would be more informative by showing a tag density plot of Insv in its different possible binding situations.
- In Fig. 2E, the elba1 mutation clearly has an impact on Elba3 recruitment yet no mention of made of this. This reinforces the need for global analysis of recruitment profiles in the different genetic contexts.
- Fig. 3B & C: to facilitate figure understanding, it would be best to keep the same order of the tracks from one figure to the next.
- In Fig. 5A: I find it surprising that the motif analysis doesn't find the γ -box (CCAAT) motif as predominant. This motif is bound by the NF-Y (dNF-Y) complex which has been shown to have pioneer factor activities which could potentially be interesting in regard to the BEN-solo proteins' motifs and functions.
- Fig. 6C & D: in the browser shots, several of the gene names are missing making the results difficult to interpret.
- Supp. Fig. 2A would be enhanced with a wild-type situation panel.

Reviewer #3 (Remarks to the Author):

Manuscript NCOMMS-19-13751-T

BEN-solo factors partition active chromatin to ensure proper gene activation in Drosophila

In this manuscript by Überschär et al. a detailed characterization of the BEN-solo proteins Insv, Elba1, Elba2 and Elba3 in transcriptional repression and chromatin insulation are performed in Drosophila embryos. To this end, the authors generated ChIP-seq data of all factors (wt and

mutants) to study chromatin binding inter-dependences, employed ChIP-nexus for precise mapping of chromatin interaction sites, determined real-time transcription changes in mutant embryos using Pro-seq, performed genetic interdependency analyses and tested insulation activity of identified sites in transgenic reporter assays. Overall, the study has been performed in a very comprehensive manner using a huge repertoire of tools, and provides important insight into the function of BEN-solo proteins in chromatin insulation to partition transcription units in active chromatin regions.

However, after careful reading, several important issues got obvious.

MAJOR POINTS:

1) One highly critical point is the different binding behaviour of the three Elba proteins in ChIP. As the authors state, this could be due to the antibodies, which is also shown in antibody stainings for Elba1 and Elba3, however not for Elba2. Quantifications are missing here (Supplementary Fig. 1A). From the pictures it also seems that Elba3 expression is reduced in elba1 mutants, and Elba1 expression is changed in elba 3 mutants. Please provide quantifications here. If this is indeed the case, this will of course also influence the ChIP experiments in the mutant backgrounds. The reviewer would also like to see antibody stainings for Elba2. One question of the reviewer is: if the authors had used tagged versions of the Elba proteins and used the same antibody, would they expect a complete overlap of Elba1, 2 and 3 peaks? Wouldn't this change the story? One suggestion would be to test the binding behaviour of tagged Elba proteins and show with the tag that they are expressed at different/equal levels and/or whether the differential binding behaviour can or cannot be captured when using the same antibody. In any case, it needs to be shown that a) Elba protein levels are unchanged in the mutant backgrounds (quantifications), and b) that the different Elba proteins indeed bind different sites when using the same antibody for ChIP experiments (for example, HA antibody for HA tagged versions of Elba). As the whole paper is about a quantitative analysis of a protein complex, this is crucial.

2) Figure 2B is completely incomprehensible. The authors should come up with a better representation of what they want to show in this Figure panel. Venn diagram in Figure S2C is much clearer.

3) The Figure legends are in general very minimalistic and do not explain very well what is shown in the images. As labelings also do not always get clear from the text, the text is sometimes confusing and needs to be read a couple of times, for example wtElba2/elba2Elba2. Please specify more clearly. Another example is Figure 4. The authors need to write either in the text or in the figure legend what the figure shows. Figure 4B is totally unclear. Is the black bar in Fig 4B "all peaks no Elba/Insv motif"? Figure 5C: what are the different phenotypic categories they are scoring? Not mentioned in the Figure, the text, the legend.

4) Sometimes the logic of performing a set of experiment is not clear and thus hard to comprehend. For example, page 9, 2nd paragraph: We next examined the 712 Elba1 peaks that remained in elba2 mutants ... " It would be helpful to introduce the paragraph with the logic of the experiment for better understanding. This whole paragraph is a bit confusing and could be better written, in particular with a better introduction.

5) The authors performed ChIP-nexus experiments to have a higher resolution of the protein binding, thereby overcoming the issue of broad ChIP-seq peaks. However, for their further analysis the authors use then the overlapping peaks from ChIP-seq and ChIP-nexus. Why? Is ChIP-nexus too noisy? The authors should show the overall overlap between ChIP-seq and ChIP-nexus. What if the authors used only the ChIP-nexus dataset for their analysis? Would it be the same as for the ChIP-seq experiment? The authors should clearly show the ChIP-nexus dataset by itself, without any overlap.

Minor points:

1. Typo on page 3, line 9: ... and but ... "
2. Page 8, line 9: the authors say "Unexpectedly, Elba3 maintains more than half of its peaks in elab1 and elba2 mutants." Why unexpectedly? If it is indeed true that half of the Elba3 binding sites are specific for Elba3, they should not dependent on Elba1 and 2.
3. Page 9, line 13: it is not 712 Elba peaks but 710 Elba peaks according to Figure S2F.
4. Why did the authors use the Drosophila genome dm3 for mapping and not dm6? Would the results be the same or similar?

In principal, the study is timely, relevant and of interest to a broad audience. However, before considering it as an article in a journal like Nature Communications, major revisions are necessary. In particular, the authors need to clarify all the issues raised, in particular the antibody issue and the quantification of Elba proteins in the respective mutant backgrounds, as this is the basis of this quantitative study.

Dear editor and reviewers,

We are submitting our much-improved manuscript “BEN-solo factors partition active chromatin to ensure proper gene activation in *Drosophila*” to be considered for publication in *Nature communications*.

We are pleased that both reviewers thought that our study is important and comprehensive. We have included a substantial amount of new results in the current version of the manuscript to address the reviewers' concerns. To streamline the ChIP analyses, we also re-organized the first part of the manuscript a bit by removing non-important figures (e.g., some genome browser screenshots) and adding new results. A few major changes are highlighted below.

1. To quantify the expression level of each Elba and Insv factor in the cognate and non-cognate mutant conditions, we have measured expression changes of *these factors* in a wild-type control and the four mutant genotypes, by using RT-qPCR. Consistent with our previous fluorescent immunostaining data (Suppl. Fig. S1A), the new results show that the mRNA level of each factor is not reduced in any of the non-cognate mutant condition (Suppl. Fig. S1A-B).

2. To prove additional Elba3 protein exist outside of the ELBA complex, we have run a sequential co-immunoprecipitation experiment (Suppl. Fig. S2E). In this experiment, we show that both the Elba1 and the Elba3 antibodies could deplete all Elba1 molecules and that the Elba3 antibody could deplete Elba3 but the Elba1 antibody could not. The results suggest that extra Elba3 molecules are present outside of the ELBA complex, supporting the conclusion that Elba3 may bind to additional genomic loci independent of ELBA.

3. We have run ChIP-qPCR to validate the ChIP-seq results using a different set of antibodies. Our results confirm that 1) Insv and ELBA co-occupy a set of genomic loci that represent strong binding of these factors to the genome; 2) Elba3-unique regions exist although these regions are weakly bound by Elba3; 3) In some genomic loci, Elba3 targets chromatin independent of the ELBA complex, while in other loci all three Elba factors rely on the formation of the ELBA complex and 4) Insv-unique regions exist.

4. We update bioinformatic analyses according to the reviewers' suggestions and also present more comprehensive comparisons between our ChIP-seq and ChIP-nexus data. Overall the ChIP-seq and the ChIP-nexus peaks overlap extensively. There is no significant difference between these two datasets regarding defining motif enrichment frequency and overlapping profiles of the binding sites between the four factors. Although the ChIP-nexus data could provide higher resolution to distinguish closely spaced binding sites, the use of the data alone is limited because the technique does not allow a meaningful negative control to be produced, perhaps due to its inherently low background.

Reviewers' comments:

Reviewer#1 (Remarks to the Author):

The manuscript submitted by Ueberschär et al. presents a genome-wide analysis of both the binding of *Drosophila* BEN-solo factors (+Elba3) and an analysis of the transcriptional repercussion of their individual mutations in an in vivo context. The authors dissect the binding interplay of the different ELBA sub-units along with Insv and show that both Elba3 and Insv have a substantial number of independent binding sites. Furthermore, they confirm their role in gene repression and go on to show a role in insulation between neighboring genes.

Chromatin insulation is a very interesting subject, and here, the authors provide a considerable amount of genome-wide data (ChIP-seq, RNA-seq, PRO-seq) that will most likely be of interest

to the Drosophila community. Additionally, the last part of the article, focusing on the insulating capacity and its developmental impact, while not particularly novel, is quite convincing. However, I must admit that I find the first half of the manuscript difficult to follow and several claims reported throughout the article are overstated. While a considerable amount of work has gone into this manuscript, unfortunately, in its current state, I cannot recommend it for publication in Nature Communications.

Here are some points that I would recommend addressing before further consideration.

Major points:

- The first half of the results section is dedicated to ELBA and Insv genome-wide binding characteristics. While this data, in a more succinct form, would be better suited for publication, what I fail to grasp is why the authors spend so much time analyzing ChIP-seq data which is clearly inferior in quality to their ChIP-Nexus results. Given the non-normalized ChIP-seq profiles compared to the ChIP-Nexus profiles (Fig. 3A), I don't see how the ChIP-seq data could prove more trustworthy and would recommend using the ChIP-Nexus data as a primary source and the ChIP-seq as a backup and not vice-versa.

We appreciate the reviewer's suggestion and we apologize for the confusion. For the reasons we present below, we can only use the ChIP-seq data to derive certain conclusions. First, our ChIP-seq analyses are high-stringent. We could obtain sufficient amount of ChIP material from all mutant condition for ChIP-seq libraries. The mutant ChIP from the cognate mutants serves the most stringent negative control (which is not always seen in the literature). Second, there is a key limitation with the ChIP-nexus data. The ChIP-nexus method shows higher resolution and specificity in the binding pattern. However, the technique does not allow to produce sufficient material from cognate mutant embryos or with a negative control antibody for analysis, presumably due to its low-background nature. This is also the case in the published ChIP-nexus datasets where no negative control was included (He Q et al, 2015; Shao W et al, 2017, two papers from Julia Zeitlinger's lab who developed the method). So in the absence of a negative control, we decided to focus on the high score peaks. To this end, we overlapped the ChIP-seq peaks (wild-type versus mutant) with the ChIP-nexus peaks in wild-type and checked for optimal Insv/Elba motif frequency and signal to background ratio.

Thus, the ChIP-nexus data alone is useful to define site asymmetry and distinguish closely spaced binding sites in high-confident peaks. However, it is not sufficient to determine factor inter-dependency in genomic binding because there is no data from the cognate and the non-cognate mutants.

In our analyses we took advantage of both the ChIP-seq and the ChIP-Nexus datasets. We first used the ChIP-seq data to elucidate co-binding of the four factors and the interdependence of factor binding because we could get good ChIP-seq signal from all wild-type and mutant conditions. We then used ChIP-Nexus to refine the binding site symmetry/asymmetry.

Furthermore, if ChIP-Nexus is supposed to provide "better specificity and resolution" as the authors claim, why are there so many less peaks called than in ChIP-seq (Fig. 1B vs Supp. Fig. 3B)? Does this mean that a large proportion of ChIP-seq peaks are false-positives?

The previous Fig. 1B (now Fig. 1C) presented the numbers of all ChIP-seq peaks, whereas the old Suppl. Fig. S3B (now removed in the current version of the manuscript) included the overlapped peaks between ChIP-nexus and ChIP-seq. Although ChIP-nexus provides better resolution for high-confident peaks, but due to the lack of a negative control (as explained above), it is more difficult to set a cut-off for low-binding sites. In order to focus on high-confident peaks to study binding site configuration, we used a very stringent cut-off to call ChIP-nexus peaks (FDR<E-10 for Elba1, Elba3 and Insv and FDR<E-5 for Elba2). The ChIP-

seq data has the cognate mutant ChIP as a negative control. We used the MACS2 default cutoff ($FDR < 1E-2$) to call confident peaks against mutant ChIP. We further chose the overlapping peaks of high-stringent ChIP-nexus peaks and ChIP-seq peaks to make sure that there are no false positive peaks called in ChIP-nexus for other analyses.

- Furthermore, in several instances, the authors mention their Elba2 ChIP-seq. While upon normalizing to mutant ChIP-seq they manage to call certain peaks (Fig. 1B), Suppl. Fig. 1B clearly shows Elba2 enrichment to be suboptimal for further analysis.

We agree that the Elba2 ChIP signal is weak due to a comparatively low affinity antibody. However, the peaks identified with this antibody show extensive overlapping with the Elba1 and Elba3 peaks. More importantly, the signals disappeared in the elba2 mutant, indicating that the Elba2 ChIP peaks we've identified are bona fide. This is further supported by the motif enrichment analysis in the new Fig. 1D that shows the motif occurrence frequency of Elba2 ChIP peaks is the same as that of Elba1 ChIP peaks and even higher than that of Elba3 and Insv especially in the low peak-score bins. Together, these results show that, despite the fact that the Elba2 ChIP signals seem to be low, the peaks we've identified represent sequences that are bound by the ELBA complex in vivo.

- One of the major findings of the paper is that both Elba3 and Insv have a large number of binding sites to which they are recruited independently of the other BEN-solo factors. Yet, the authors fail to delve into the description of the particularities of these sites beyond genetic localization and motif representation (Suppl. Fig. 3F). It would have been interesting to know if different categories of genes are regulated by solo rather than by complex binding. Or do solo sites have a different effect on gene regulation than complex sites (according to Suppl. Fig. 4, independent Insv sites seem to upregulate approximately as many genes as downregulate them). Or yet, do these proteins co-localize with different sets of proteins depending on if they bind solo or in a complex?

This is a good suggestion. In our current version, we have made new analyses for 5 different categories of binding sites including the 4-factor overlapping regions, the Elba123 overlapping but no Insv binding regions, the Elba3 and Insv overlapping regions, the Elba3-unique and the Insv-unique regions, and we assessed: 1) ChIP-seq read coverage (Suppl. Fig. S1D); 2) Insv/Elba motif enrichment (Suppl. Fig. S1E); 3) gene expression change associated with these peaks (Fig. 5C); 4) Other insulator site enrichment (Suppl. Fig. S6A). Below are the main observations.

1. The Elba3-unique sites (that do not overlap with any other factor binding sites) have lowest peak signal and motif enrichment (Suppl. Fig S1D-E). The associated genes are not downregulated in the elba3 mutant (Fig. S5C), so these sites are likely non-functional.

2. The expression of the genes bound by Insv and Elba3 but not by Elba1 and Elba2 do not show any change in the elba3 mutant. There is mild upregulation in the insv mutant but the change is not statistically significant. This suggests that the function of Elba3 or Insv is dispensable for the regulation of these genes (Suppl. Fig. S5C).

3. The genes associated with the Insv-unique sites tend to be upregulated in the insv mutant but again the change is not statistically significant, suggesting Insv binding is dispensable for the regulation of these genes in this stage (Suppl. Fig. S5C).

4. The Elba3-unique or the Insv-unique sites do not show higher enrichment of other Insulator sites including those for CP190, BEAF32, and GAF. Instead, the 4-factor overlapping sites show highest enrichment of the motifs for these three insulators (Suppl. Fig. S6A).

Based on these observations, we conclude that the genes associated with binding peaks of all four factors are functional targets regulated by ELBA and Insv in the early embryo.

- The same case could be made about the focus on binding sites containing motifs. While I

understand that the motifs offer some confirmation as to the veracity of these binding sites, the fact that 71 and 75% for of Elba3 and Insv binding sites (Supp. Fig. 3A), respectively, show no typical motif enrichment is both puzzling and fascinating. Yet, no mention is made of this.

There are a set of ChIP peaks that do not contain the Insv/Elba motifs. These regions could be genomic loci indirectly bound by Insv or ELBA. In the new Fig. 1D, we show that the motif enrichment anti-correlates with the peak scores, a typical pattern of good ChIP-seq data. For example, 60% of high-score peaks contain the Insv/Elba motifs but the motif enrichment frequency declines with the peak scores to as low as <10% (Insv) or <25% (Elba3). When we analyzed gene expression changes associated with the peaks with or without the motifs, we detected higher upregulation in the genes with the motifs than those without the motifs. This result suggests that regulation by ELBA and Insv is essential for proper expression of their direct targets but is dispensable for that of the genes indirectly bound by ELBA and Insv.

- A certain number of claims are made by the authors to which I find the data to prove them insufficient.

Examples:

- in elba2 mutants, Elb1, Elba3 and Insv are unlikely to form a complex.

We now changed the statement to, “This result argues that Insv does not contribute to Elba1 and Elba3 binding to these new genomic loci (Suppl. Fig. S2G).”

- Insv is involved in recruiting Elba3 (especially since in the next paragraph, the authors study Elba3 recruitment in a cell-type that does not express Insv).

We now changed the statement to “This suggests that Insv contributes to or enhances Elba3 binding in some but not all Elba1/2-independent Elba3 loci. “

- ELBA and Insv only repress transcription and the down-regulated genes are due to “random overlapping” (this is particularly hard to assess since there is no further analysis of these genes/binding sites).

We now changed the statement to “the down-regulated genes show less overlapping (Suppl. Fig. S5B), indicating these genes are not co-regulated direct targets of the Elba factors and Insv.”

- The insulation function of Insv-bound sites is restricted to “certain pairs of genes” ◊ too vague.

We now changed the statement to “in many individual loci, we observed a similar reduction of expression difference between Insv-bound neighbor promoters in the insv mutant (Fig. 6C-E), suggesting that the insulation function of Insv-bound sites is present in early embryos but the impact may not be as global as that of ELBA-bound sites.”

Minor points:

- The manuscript contains a large number of typos and grammatical errors which sometimes make it hard to follow (such as Mdg4/Mdj4, cortex/contexts, etc...).

We appreciate the reviewer’s careful reading. Now we have done more rounds of proof reading and hopefully the number of typos and errors are minimized.

- I’m not quite clear on how Supp. Fig. 1A shows that the “peak expression” levels of the ELBA factors are during the blastoderm stage of the embryos.

We apologize for the confusion. The images in Suppl. Fig. S1A show the immunostaining of Elba1 and Elba3 in blastoderm stage of embryos. The statement of “peak expression” is based on our previous publication (Dai Q et al. 2015), where we showed that Elba1 and Elba3 are highly expressed in blastoderm stages and then disappear for the rest of developmental stages. We now revised the relevant description in the text.

- While the immunofluorescence images in Supp. Fig. 1A are appreciable, they do not allow for precise quantification of the effects of mutating the Ben-solo + Elba3 genes on each other. Therefore, a quantified Western-blot, or, if no antibodies of sufficient quality are available, qRT-PCR experiments of all analyzed factors in all mutant conditions should be presented in order to truly appreciate the effect of the mutations. This would also allow to show the effects of Elba2 mutation.

Following this suggestion, we first tried western-blot to measure the protein level of each factor in the five genotypes. However, none of the antibodies against these four factors has worked on western-blot with embryo extract. In particular, all four factors are about 40KD and co-migrate with yolk protein. As there are massive amounts of yolk protein in early embryos, this makes detection of the four proteins by westerns essentially impossible. We then tried to immunoprecipitate the proteins and detect them in Western-Blot. Although the antibodies show protein bands at the expected position and the bands disappeared in the cognate mutants, indicative of high specificity of the IPs, the pulldown efficiency was somehow variable and cannot be quantified as no loading control can be used in this type of assay. We include the result from one exemplary experiment in the end of the letter to show to the reviewers. We will be willing to include it in the supplemental data if necessary. In this particular experiment, there is a slight reduction of Elba1 and Elba2 in the elba3 mutant. However, in other experiments, we also saw no reduction. Nevertheless, in all cases, Elba factors never disappeared in the non-cognate elba mutants and were always gone in the cognate mutants. Even if there was reduction in levels, it could not be the reason for nearly complete elimination of all Elba1 and Elba2 ChIP peaks in the elba3 mutant.

We have also performed RT-qPCR and included the results in the new supplemental Fig. S1B. In each of the cognate mutants, the wild-type mRNAs of each gene are nearly undetectable, as expected. In each of the non-cognate mutants, the wild-type mRNAs of each gene are not reduced. Combining this result with the immunostaining data (Suppl. Fig.S1A), we conclude that the reduction or depletion of Elba factor binding in the non-cognate elba mutants is not due to reduced expression of these factors, but due to the loss of ELBA binding in their targets.

- If my understanding is correct, Supp. Fig. 1C shows the genetic positioning of Elba3 with and without Insv. However, in this figure when adding both amounts (3729+2555) we fail to reach the 6525 called peaks mentioned in the text. The same is true for Insv.

Thanks for the reviewer’s careful reading. To compare unique and common peaks among different peak sets, we merged the peaks from the relevant peak sets. Note that we used unique merged peak ids to count #peaks after merging, so the final merged #peaks are always smaller than the total input peaks from different sets. Also note that even for the same peak set, more than 2 peaks could be merged to one if their distance less than a defined maximum distance (we set <100nt). Furthermore, depending on how many peak sets involved in merging, the number of final merged peaks can be slightly different.

Take the example that you mentioned, Elba3 has 6525 unique merged peaks in Figure 1C by merging 4 wt against their cognate mutant peak sets: wtElba1/elba1Elba1, wtElba2/elba2Elba2, wtElba3/elba3Elba3, wtInsv/inslInsv. As the analysis for the previous Suppl Fig.S1C was done together with main Figure 2 involving 16 peak set merging, the final unique merged peaks for Elba3 is 6284 (3729+2555). The same for Insv.

Note that in our new version, we have added many new analyses and re-arranged our figures to better present our results. The previous Suppl Fig.S1C is not included in the current manuscript.

- Fig. 1D, Elba2 data is missing.

Now in the new data Fig. 1E, we added the Elba2 data plot. Despite the Elba2 ChIP signal is overall lower compared to that of the other three factors, the pattern of motif distribution in Elba2 ChIP peaks is the same as in the other Elba2 factors: the peaks containing symmetric motif and asymmetric motifs have similar signal intensity, suggesting Elba2 (similar to Elba1 and Elba3) associates with both types of motifs in vivo.

- While I don't disagree with the statement, I believe simple tag-density plots showing Elba3 ChIP-seq signal enrichment in the different types of scenarios (with or without Elba1/2 or Insv) would be easier to understand and perhaps more informative than the cumulative distributions of peak scores (ie Supp. Fig. 2E).

- Similarly, rather than showing a browser shot (with a tiny y-axis compared to other browser shots, possibly indicating background noise), Supp. Fig. 3E would be more informative by showing a tag density plot of Insv in its different possible binding situations.

As suggested, we have added the following tag density plots

- 1) Suppl Fig. S2D: for Elba3 binding (with or without Elba1/2 or Insv)*
- 2) Suppl Fig. S2H: for Elba1 binding (wt and in elba2 mutant as well as Elba3 in elba2 mutant)*
- 3) New figure – Suppl Fig. S1D: 5 subsets*

- In Fig. 2E, the elba1 mutation clearly has an impact on Elba3 recruitment yet no mention of made of this. This reinforces the need for global analysis of recruitment profiles in the different genetic contexts.

We now commented on this observation, “Notably, the enrichment of Elba3 appears lower in the elba1 and the elba2 mutants, suggesting the presence of Elba1 and Elba2 is dispensable for but could stabilize Elba3 binding.”

- Fig. 3B & C: to facilitate figure understanding, it would be best to keep the same order of the tracks from one figure to the next.

Yes we agree. We have corrected Figure 3C to make the track orders the consistent with Figure 3B.

- In Fig. 5A: I find it surprising that the motif analysis doesn't find the y-box (CCAAT) motif as predominant. This motif is bound by the NF-Y (dNF-Y) complex which has been shown to have pioneer factor activities which could potentially be interesting in regard to the BEN-solo proteins' motifs and functions.

We were also aware of the Elba/Insv motifs CCAATAAG and CCAATTGG contain the sequence of the Y-box CCAAT. However, in the motif discovery, we did not detect CCAATNNN alone as a predominant motif. The motifs from MEME always contain CCAATAAG, CCAATTGG or their variants.

- Fig. 6C & D: in the browser shots, several of the gene names are missing making the results difficult to interpret.

Thanks for the reviewer's careful reading. We have checked and added all the missing gene names in the browser screenshots in all related figures

- Supp. Fig. 2A would be enhanced with a wild-type situation panel.

Thanks for the suggestion. We have added the peak numbers from wt.

Reviewer #3 (Remarks to the Author):

Manuscript NCOMMS-19-13751-T

BEN-solo factors partition active chromatin to ensure proper gene activation in *Drosophila*

In this manuscript by Überschär et al. a detailed characterization of the BEN-solo proteins Insv, Elba1, Elba2 and Elba3 in transcriptional repression and chromatin insulation are performed in *Drosophila* embryos. To this end, the authors generated ChIP-seq data of all factors (wt and mutants) to study chromatin binding inter-dependences, employed ChIP-nexus for precise mapping of chromatin interaction sites, determined real-time transcription changes in mutant embryos using Pro-seq, performed genetic interdependency analyses and tested insulation activity of identified sites in transgenic reporter assays. Overall, the study has been performed in a very comprehensive manner using a huge repertoire of tools, and provides important insight into the function of BEN-solo proteins in chromatin insulation to partition transcription units in active chromatin regions.

However, after careful reading, several important issues got obvious.

MAJOR POINTS:

1) One highly critical point is the different binding behaviour of the three Elba proteins in ChIP. As the authors state, this could be due to the antibodies, which is also shown in antibody stainings for Elba1 and Elba3, however not for Elba2. Quantifications are missing here (Supplementary Fig. 1A). From the pictures it also seems that Elba3 expression is reduced in elba1 mutants, and Elba1 expression is changed in elba 3 mutants. Please provide quantifications here. If this is indeed the case, this will of course also influence the ChIP experiments in the mutant backgrounds. The reviewer would also like to see antibody stainings for Elba2. One question of the reviewer is: if the authors had used tagged versions of the Elba proteins and used the same antibody, would they expect a complete overlap of Elba1, 2 and 3 peaks? Wouldn't this change the story? One suggestion would be to test the binding behaviour of tagged Elba proteins and show with the tag that they are expressed at different/equal levels and/or whether the differential binding behaviour can or cannot be captured when using the same antibody. In any case, it needs to be shown that a) Elba protein levels are unchanged in the mutant backgrounds (quantifications), and b) that the different Elba proteins indeed bind different sites when using the same antibody for ChIP experiments (for example, HA antibody for HA tagged versions of Elba). As the whole paper is about a quantitative analysis of a protein complex, this is crucial.

Regarding the comment a): we first tried western-blot to measure the protein level of each factor in the five genotypes. However, none of the antibodies against these four factors has worked on western-blot with embryo extract. In particular, all four factors are about 40KD and

co-migrate with yolk protein. As there are massive amounts of yolk protein in early embryos, this makes detection of the four proteins by westerns essentially impossible. We then tried to immunoprecipitate the proteins and detect them in Western-Blot. Although the antibodies show protein bands at the expected position and the bands disappeared in the cognate mutants, indicative of high specificity of the IPs, the pulldown efficiency was somehow variable and cannot be quantified as no loading control can be used in this type of assay. We include the result from one exemplary experiment in the end of the letter to show to the reviewers. We will be willing to include it in the supplemental data if necessary. In this particular experiment, there is a slight reduction of Elba1 and Elba2 in the elba3 mutant. However, in other experiments, we also saw no reduction. Nevertheless, in all cases, Elba factors never disappeared in the non-cognate elba mutants and were always gone in the cognate mutants. Even if there was reduction in levels, it could not be the reason for nearly complete elimination of all Elba1 and Elba2 ChIP peaks in the elba3 mutant.

We have also performed RT-qPCR and included the results in the new supplemental Fig. S1B. In each of the cognate mutants, the wild-type mRNAs of each gene are nearly undetectable, as expected. In each of the non-cognate mutants, the wild-type mRNAs of each gene are not reduced. Combining this result with the immunostaining data (Suppl. Fig.S1A), we conclude that the reduction or depletion of Elba factor binding in the non-cognate elba mutants is not due to reduced expression of these factors, but due to the loss of ELBA binding in their targets.

Regarding the comment b): if we understand it right, the reviewer suggests us to tag the endogenous loci for all the four factors with a common tag and re-do the ChIP-seq experiments. For the reasons listed below, we don't think it is necessary and helpful to do such experiments.

1) It is the first time that the ChIP-seq analyses were done for the Elba family of proteins. We compared all possible negative controls (IgG, Input and mutant ChIP) and used mutant ChIP as the most stringent control for peak calling. We argue that our ChIP analyses are original and of high stringency.

2) Our ChIP-seq analysis revealed an unexpected property of Elba3 binding independent of the Elba complex. In the wild-type condition, there are more ChIP loci identified for Elba3 than Elba1 and Elba2 (after comparing peaks from wild-type normalized to cognate-mutant ChIP, Fig 1 B-C). This could result from difference on antibody affinity. To address this concern, we took two strategies. First, we performed ChIP-qPCR by using a different set of antibodies and confirmed the ChIP-seq result-the Elba3-unique sites are not bound by Elba1 and Elba2 (Suppl. Fig S2A-D). Second, we performed a sequential co-immunoprecipitation experiment where we use the Elba1 and the Elba3 antibodies to deplete the ELBA complex and then check in the left-over protein extract for the presence or absence of Elba1 and Elba3 (Suppl. Fig S2E). We show the Elba3 antibody depletes nearly all the Elba1 and Elba3 proteins while the Elba1 antibody could only deplete all the Elba1 proteins and leave additional Elba3 protein molecules in the depleted extract. This result suggests that the Elba3 protein could exist beyond the ELBA complex, supporting the conclusion that Elba3 can target other genomic loci independent of ELBA.

3) Another strength of our ChIP analyses is that we performed ChIP-seq from all five genetic conditions including wild-type, the cognate and the non-cognate mutants (illustrated in Fig. 1A). In the non-cognate mutant condition, we examined interdependency between the factors. Elba3 retains binding to about half of its sites in the elba1 and the elba2 mutants (Fig 2 and Fig S3), supporting the finding that Elba3 is able to target chromatin independent of the ELBA complex.

4) We have found that tagging Elba1 interferes with its activity. We have tried V5 and VP16 at the N- or C-terminus in overexpression experiments, but none of them worked. This raises a concern that the function or protein configuration of Elba1 may be easily disrupted by addition

of a tag, and this in turn could alter the behavior of the ELBA complex in vivo. Another concern is that even though two different proteins carry the same tag, the antibody against the tag need not recognize the two tagged proteins with the same efficiency. In one case, the tag might readily accessible, while in the other case it might not. Moreover, the efficiency of recognition could differ when the tagged protein is in a complex and when it is not. Likewise, if proteins in the immediate vicinity alter accessibility, a tagged protein bound at one site need not be recognized by the anti-tag antibody with the same efficiency as it is recognized at another genomic location. So in addition to the possibility of perturbing ELBA complex formation, its not clear that adding tags to the three Elba proteins would significantly improve the reliability of our ChIPs. More importantly, the tags are unlikely to alter our main findings/conclusions. In looking at our data, the most important point to be drawn from the ChIPs-namely that the Elba proteins Elba1 and Elba2 bind to DNA as part of a complex with Elba3-will be the same. Similarly our experiments showing that this complex is the relevant biological entity will also remain the same. The fact that Elba3 can associate with chromatin in the elba1 or elba2 mutants will also remain the same. Probably the only difference will be in the number of peaks identified in Elba2 ChIPs (assuming that the tag does not interfere with ELBA complex assembly/function).

5) *To duplicate the ChIP-seq experiments using a tag antibody, we would need to tag each factor in 5 genotypes (wt, elba1, elba2, elba3 and insv) because these four factors locate closely in one chromosome arm and thus recombination between the loci will be nearly impossible. For each one, that's five stocks that need to be constructed and 4x5 ChIPs that need to be performed. So it is an experiment that is not practical and for only an unpredictable return – in the best case a better quantitation if we assume that the recovery of tagged Elba1, Elba2, Elba3 and Insv is unbiased, the same result as what we got with the current antibodies or a confusing situation if the tag changes the properties of one or more proteins.*

2) Figure 2B is completely incomprehensible. The authors should come up with a better representation of what they want to show in this Figure panel. Venn diagram in Figure S2C is much clearer.

We agree that this graph is confusing and have now replaced it with the previous Suppl. Fig. S2C. The difference is that Fig. S2C does not contain wtElba1 and wtElba2 peaks. However, our point that a fraction of Elba3 sites are maintained in the elba1/2 mutants and are thus independent of Elba1 and Elba2 is still clear with this figure.

3) The Figure legends are in general very minimalistic and do not explain very well what is shown in the images. As labelings also do not always get clear from the text, the text is sometimes confusing and needs to be read a couple of times, for example wtElba2/elba2Elba2. Please specify more clearly. Another example is Figure 4. The authors need to write either in the text or in the figure legend what the figure shows. Figure 4B is totally unclear. Is the black bar in Fig 4B “all peaks no Elba/Insv motif”? Figure 5C: what are the different phenotypic categories they are scoring? Not mentioned in the Figure, the text, the legend.

We apologize for the confusion. Now we illustrate how the datasets are named (Fig. 1A). We have added necessary information to the Figure legends.

4) Sometimes the logic of performing a set of experiment is not clear and thus hard to comprehend. For example, page 9, 2nd paragraph: We next examined the 712 Elba1 peaks that remained in elba2 mutants ... “ It would be helpful to introduce the paragraph with the logic of the experiment for better understanding. This whole paragraph is a bit confusing and could be better written, in particular with a better introduction.

We thank for the reviewer's careful reading and apologize for any confusion in the manuscript. Now we have made substantial amount of modifications on the figures and texts and hope the writing has been improved significantly.

5) The authors performed ChIP-nexus experiments to have a higher resolution of the protein binding, thereby overcoming the issue of broad ChIP-seq peaks. However, for their further analysis the authors use then the overlapping peaks from ChIP-seq and ChIP-nexus. Why? Is ChIP-nexus too noisy? The authors should show the overall overlap between ChIP-seq and ChIP-nexus. What if the authors used only the ChIP-nexus dataset for their analysis? Would it be the same as for the ChIP-seq experiment? The authors should clearly show the ChIP-nexus dataset by itself, without any overlap.

The ChIP-nexus method shows higher resolution and specificity of binding. However, the ChIP-nexus method does not allow sufficient material from cognate mutant embryos or with a negative control antibody to be produced. This is presumably due to the low-background nature of this method. Thus, the ChIP-nexus data does not have a negative control. This was also the case in the previously published ChIP-nexus studies (He Q et al, 2015; Shao W et al, 2017, two papers from Julia Zeitlinger's lab who developed the method). So in the absence of a negative control, we decided to focus on high-confident peaks with very stringent cutoff (FDR < E-10 for Elba1, Elba3 and Insv, FDR < E-5 for Elba2). After applying this cutoff, the ChIP-nexus peaks are mostly covered by the corresponding ChIP-seq peaks (which was first normalized to cognate mutant ChIP) (Suppl. Fig. S4A-C). The reason to use the overlapping peaks of ChIP-seq and ChIP-nexus is simply to take advantage of the narrow peaks defined by ChIP-nexus but also to minimise false positive signal resulted from lack of a negative control in ChIP-nexus as the ChIP-seq peaks are called against mutant ChIP.

Following the reviewer's suggestion, we now present more comparison between ChIP-seq and ChIP-nexus (Suppl. Fig. S4). The motif enrichment frequency is similar between these two datasets and slightly increased in the overlapping peaks (Suppl. Fig. S4A, S4C). The ChIP-nexus data gives better resolution of factor overlapping within a very short range (10 nts) than ChIP-seq (Suppl. Fig. S4D). However, when allowing larger range, there is no difference between the two methods to define factor overlapping (Suppl. Fig. S4D).

Minor points:

1. Typo on page 3, line 9: ... and but ... "

We appreciate the reviewer's careful reading and have now minimized the typos in the text.

2. Page 8, line 9: the authors say "Unexpectedly, Elba3 maintains more than half of its peaks in elab1 and elba2 mutants." Why unexpectedly? If it is indeed true that half of the Elba3 binding sites are specific for Elba3, they should not dependent on Elba1 and 2.

This observation is unexpected, because it is in contrast to the previously published information where Elba3 just acted as an adaptor protein that bridges the two DNA binding subunits, Elba1 and Elba2, of the trimeric complex ELBA. It was not expected bind to the genome without Elba1 and Elba2. However, it is not unexpected based on our ChIP-seq analyses where Elba3 shows uniquely-bound regions, presented in the first part of the manuscript.

3. Page 9, line 13: it is not 712 Elba peaks but 710 Elba peaks according to Figure S2F.

We appreciate the reviewer's careful reading. There are a total of 712 Elba1 peaks in the elba2 mutant. Two of these peaks that we did not label are shared by elba2Elba1 and wtElba1 but

not in ebla2Elba3. To make it clear, we now update the figure by adding the label of these two peaks in Supplemental Figure S3F.

4. Why did the authors use the Drosophila genome dm3 for mapping and not dm6? Would the results be the same or similar?

This study started several years ago before the dm6 is available. To keep our new sequencing data consistent to the older data, we decided to keep dm3 as the reference genome. We believe that the results and conclusions are the same or very similar using two versions of reference genomes. We have deposited the raw data (fastq files) for all the sequencing data we generated to the GEO, so researchers can easily map them to dm6 to their needs.

In principal, the study is timely, relevant and of interest to a broad audience. However, before considering it as an article in a journal like Nature Communications, major revisions are necessary. In particular, the authors need to clarify all the issues raised, in particular the antibody issue and the quantification of Elba proteins in the respective mutant backgrounds, as this is the basis of this quantitative study.

We appreciate that the reviewer thinks our study is timely, relevant and have broad interest. We hope that in the current version of the manuscript, we have addressed the major concerns.

REVIEWERS' COMMENTS:

Reviewer #1 (Remarks to the Author):

I am quite pleased to say that the revised version of the paper submitted by Ueberschär et al. has considerably improved since its original submission to Nature Communications.

The authors addressed most of my previous concerns by either offering more detailed explanations or by adding new data that goes a long way into clarifying the presented story.

I would particularly like to commend the authors on their elegant experiment leading to Figure S2E. Furthermore, they did a very good job of disentangling the ChIP-seq vs. ChIP-nexus data and the current version flows much better and is much more comprehensible.

However, a few minor points would benefit from a little clarification before publication:

- The qRT-PCR experiment (Fig. S1B) is quite informative and reassuring as far as the fact the none of the proteins other than the KO'd ones are downregulated. However, the authors don't address the fact that some of them are quite strongly upregulated (> 3-fold) in some of the conditions. This seems to be true at the protein level as well according to the WB attached to the author's letter. Could this strong upregulation not have an impact on binding? Do the upregulated proteins bind to additional sites? I believe this is of certain importance given that between lines 317 and 328 the authors observe "shifting" of Elba1 binding sites from the WT positions to sites previously bound by Elba3.

- Statistics are missing in Figure S1C (the histogram). It isn't clear if the motif enrichment at the protein binding sites is significantly stronger than at Input or IgG sites.

- I was confused by line 234/235: "The Elba3 and Insv co-binding regions have similar average wt/mutant coverage to the ones co-bound by the three Elba factors". Are the authors not referring to the the grey vs the blue lines (which are quite different)?

Overall, I now believe that the manuscript is suitable for publication pending clarification of these minor details.

Reviewer #3 (Remarks to the Author):

Manuscript NCOMMS-19-13751-A

BEN-solo factors partition active chromatin to ensure proper gene activation in Drosophila

The manuscript has been significantly improved, the authors have put a significant amount of effort into the revisions to address the concerns of this reviewer, which is highly appreciated. The newly added experiments address most of the points raised by this reviewer, in particular the attempts to quantify expression levels Elba and Insv factors in the cognate and non-cognate mutant conditions using different approaches (Western-Blot, IP, by -qPCR). With the added information, it is now also clearer why the authors do not solely rely on the ChIP-nexus results but use a combination of ChIP-seq and ChIP-nexus studies. The comprehensiveness of the manuscript has also increased substantially due to better illustrations and explanations. Thus, I recommend the manuscript without any further experiments in Nature Communications.

Dear editor and reviewers,

We are pleased that both reviewers thought that the last version of our manuscript has been significantly improved. In the current version, we have formatted the text, figures, supplementary data files to meet the requirement by Nature Communications. We have also updated the text to correct and clarify the points raised by reviewer #1.

Below are Point-to-point responses to REVIEWERS' COMMENTS:

Reviewer #1 (Remarks to the Author):

I am quite pleased to say that the revised version of the paper submitted by Ueberschär et al. has considerably improved since its original submission to Nature Communications. The authors addressed most of my previous concerns by either offering more detailed explanations or by adding new data that goes a long way into clarifying the presented story. I would particularly like to commend the authors on their elegant experiment leading to Figure S2E. Furthermore, they did a very good job of disentangling the ChIP-seq vs. ChIP-nexus data and the current version flows much better and is much more comprehensible.

We appreciate the reviewer's positive feedback.

However, a few minor points would benefit from a little clarification before publication:
- The qRT-PCR experiment (Fig. S1B) is quite informative and reassuring as far as the fact the none of the proteins other than the KO'd ones are downregulated. However, the authors don't address the fact that some of them are quite strongly upregulated (> 3-fold) in some of the conditions. This seems to be true at the protein level as well according to the WB attached to the author's letter. Could this strong upregulation not have an impact on binding? Do the upregulated proteins bind to additional sites? I believe this is of certain importance given that between lines 317 and 328 the authors observe "shifting" of Elba1 binding sites from the WT positions to sites previously bound by Elba3.

This is an interesting point. We also realized an increase of *elba1* and *elba3* mRNA levels in the *elba2* mutant as well as a milder increase of *elba1* and *elba2* mRNAs in the *elba3* mutant. This might be due to genetic compensation, a phenomenon becoming more appreciated recently where gene paralogues compensate each other's expression when one factor is depleted in the cell. We have added such speculation in the main text.

Because the measurement of protein levels using immunoprecipitation followed by western blot (the image attached in the previous rebuttal letter) was not quantitative, we are uncertain whether an increase of their mRNAs leads to an increase of these protein levels. Moreover, even if there was an increase of their protein levels, Elba1 and Elba2 binding to the genome is eliminated in the *elba3* mutant (Figure 2A). This means an increase of the expression of these factors cannot increase the number of their binding sites. Instead, Elba1 and Elba2 rely on the formation of the ELBA complex with Elba3 to target their genomic sites. Similarly, but less dramatically, in the *elba2* mutant the total number of Elba1 peaks is mostly gone (decreased from 3151 in *wt* to 712 in *elba2*). The gained Elba1 peaks in the *elba2* mutant that are not present in *wt* overlap with Elba3 peaks (Supplementary Figure 3F), so we think Elba1 binding to these sites is more likely due to the recruitment by Elba3 and less likely due to the increased Elba1 protein level, although we cannot rule the latter possibility completely.

- Statistics are missing in Figure S1C (the histogram). It isn't clear if the motif enrichment at the protein binding sites is significantly stronger than at Input or IgG sites.

Thanks for noticing this. We now add the statistics calculated using fisher's exact test (two-sided). * $p < 0.05$, * $p < 0.005$, *** $p < 0.0005$. These show that the motif enrichment of the

wt/mutant peaks is significantly higher than that of the peaks called against Input or IgG for Elba1, Elba2 and Insv. For Elba3, the wt/mutant peaks have significantly higher motif enrichment than those called against IgG but not those called against Input. The information has been included in the corresponding figure legend.

- I was confused by line 234/235: "The Elba3 and Insv co-binding regions have similar average wt/mutant coverage to the ones co-bound by the three Elba factors". Are the authors not referring to the the grey vs the blue lines (which are quite different)?

We appreciate the reviewer's careful reading. This was indeed a mistake we made. During the last round of revision, the graph was modified once to make the color code consistent with the graph in Supplementary Figure 5C. Then there was some confusion between the modified and unmodified versions. We now have corrected the mistake in the text, "The four-factor overlapping sites show the highest average coverage and motif enrichment, followed by those co-bound by the Elba factors and those by Elba3 and Insv. The Insv-unique and the Elba3-unique sites have the lowest signal."

Overall, I now believe that the manuscript is suitable for publication pending clarification of these minor details.

Reviewer #3 (Remarks to the Author):

Manuscript NCOMMS-19-13751-A

BEN-solo factors partition active chromatin to ensure proper gene activation in Drosophila

The manuscript has been significantly improved, the authors have put a significant amount of effort into the revisions to address the concerns of this reviewer, which is highly appreciated. The newly added experiments address most of the points raised by this reviewer, in particular the attempts to quantify expression levels Elba and Insv factors in the cognate and non-cognate mutant conditions using different approaches (Western-Blot, IP, by -qPCR). With the added information, it is now also clearer why the authors do not solely rely on the ChIP-nexus results but use a combination of ChIP-seq and ChIP-nexus studies. The comprehensiveness of the manuscript has also increased substantially due to better illustrations and explanations. Thus, I recommend the manuscript without any further experiments in Nature Communications.

We appreciate the reviewer's positive feedback.